# Chaperone activation and client binding of a 2-cysteine peroxiredoxin

Filipa Teixeira[1,2,3,4], Eric Tse [5], Helena Castro[2,3], Karl A.T. Makepeace[6,7], Ben A. Meinen[1,8], Christoph H. Borchers[6,7,9,10], Leslie B. Poole[11], James C. Bardwell[1,8], Ana M. Tomás[2,3,4], Daniel R. Southworth[5] & Ursula Jakob[1]

Many 2-Cys-peroxiredoxins (2-Cys-Prxs) are dual-function proteins, either acting as peroxidases under non-stress conditions or as chaperones during stress. The mechanism by which 2-Cys-Prxs switch functions remains to be defined. Our work focuses on *Leishmania infantum* mitochondrial 2-Cys-Prx, whose reduced, decameric subpopulation adopts chaperone function during heat shock, an activity that facilitates the transition from insects to warm-blooded host environments. Here, we have solved the cryo-EM structure of mTXNPx in complex with a thermally unfolded client protein, and revealed that the flexible N-termini of mTXNPx form a well-resolved central belt that contacts and encapsulates the unstructured client protein in the center of the decamer ring. In vivo and in vitro cross-linking studies provide further support for these interactions, and demonstrate that mTXNPx decamers undergo temperature-dependent structural rearrangements specifically at the dimer-dimer interfaces. These structural changes appear crucial for exposing chaperone-client binding sites that are buried in the peroxidase-active protein.

[1] Department of Molecular, Cellular and Developmental, University of Michigan, Ann Arbor 48109-1085 MI, USA. [2] i3S - Instituto de Investigação e Inovação em Saúde, Universidade do Porto, Porto 4200-135, Portugal. [3] IBMC - Instituto de Biologia Molecular e Celular, Universidade do Porto, Porto 4050-313, Portugal. [4] ICBAS – Instituto de Ciências Biomédicas Abel Salazar, Universidade do Porto, Porto 4050-313, Portugal. [5] Department of Biochemistry and Biophysics, Institute for Neurodegenerative Diseases, University of California, San Francisco 94158 CA, USA. [6] Department of Biochemistry and Microbiology, University of Victoria, Victoria V8P 5C2 BC, Canada. [7] Genome British Columbia Proteomics Centre, University of Victoria, Victoria V8Z 7X8 BC, Canada. [8] Howard Hughes Medical Institute, Ann Arbor 48109-1085 MI, USA. [9] Gerald Bronfman Department of Oncology, Jewish General Hospital, Montreal H4A 3T2 QC, Canada. [10] Proteomics Centre, Segal Cancer Centre, Lady Davis Institute, Jewish General Hospital, Montreal H3T 1E2 QC, Canada. [11] Department of Biochemistry, Wake Forest School of Medicine, Winston-Salem 27157 NC, USA. These authors contributed equally: Filipa Teixeira, Eric Tse. Correspondence and requests for materials should be addressed to D.R.S. (email: daniel.southworth@ucsf.edu) or to U.J. (email: ujakob@umich.edu)

Peroxiredoxins are ubiquitous, highly abundant proteins found in every biological kingdom[1]. Best known for their ability to detoxify a variety of different peroxides, peroxiredoxins act as general antioxidants, sophisticated regulators of peroxide-dependent cell signaling pathways and thiol oxidases[2,3]. The catalytic activity of 2-Cys-peroxiredoxins (from hereon abbreviated as Prx), which comprise the Prx1 family[4], is mediated by the active site peroxidatic cysteine $C_p$, which reacts with peroxide and related oxidants, and undergoes reversible sulfenic acid formation[5]. Attack by a second cysteine that is located in the other subunit of the Prx-dimer leads to the formation of a disulfide bond, which is typically resolved by the thioredoxin system to enable another catalytic cycle[6].

Prxs have long been known to undergo major reversible changes in their quaternary structure during redox cycling. The basic structural unit of Prx is a homo-dimer, in which two subunits are organized in a head-to-tail orientation, stabilized through the antiparallel arrangement of two β-strands (i.e., B-type or β-sheet based interface)[4]. In the reduced state, most Prx1 family members associate into donut-shaped ring-like decamers. The active site cysteine-containing $C_p$-loop-helix adopts a closed conformation through an elaborate network of electrostatic interactions, thereby exposing critical aromatic amino acids that pack tightly against the other dimer, stabilizing the dimer–dimer interface (i.e., A-type or alternate interface)[5]. Upon oxidation of the active site cysteine, the $C_p$-loop-helix transitions into a more open conformation (i.e., unfolded state), which leads to the dissociation into oxidized dimers[5,7]. Overoxidation of the active site cysteine to sulfinic acid has been shown to cause the formation of even higher molecular weight oligomeric structures, including filamentous or spherical structures[8–10]. These higher oligomeric structures were reported to take on a peroxidase-independent second function as molecular chaperones, which protect cells against stress-induced protein unfolding[8], and serve as an integrated member of the eukaryotic proteostasis network during distinct stress conditions[11]. Other conditions that trigger the functional switch from a peroxidase to a chaperone through changes in the oligomeric status include exposure to low pH[12] or phosphorylation events in the Cp-loop-helix[13].

We recently reported that the mitochondrial 2-Cys Prx of *Leishmania infantum* (mTXNPx, Prx1m) also adopts two functions, as a peroxidase and as a molecular chaperone[14,15]. However, in contrast to previous studies with cytosolic Prx from yeast or mammalian cells[8,16,17], we found that neither overoxidation of the active site cysteine nor the formation of higher oligomeric structures were necessary to convert the peroxidase into a chaperone[14]. In fact, our data revealed that reduced mTXNPx decamers alone serve as an effective chaperone reservoir when exposed to physiologically relevant heat shock conditions[14,15]. Once activated by elevated temperatures, reduced mTXNPx decamers protect a range of proteins from heat-induced aggregation both in vitro and in vivo[14,15]. Two separate studies came to similar conclusions and showed that both plant C2C-Prx1 as well as mitochondrial Prx from the anaerobic archaeon *Thermococcus kodakaraensis* serve as molecular chaperones specifically under heat shock conditions[18,19]. Upon return to non-heat stress conditions, mTXNPx then transfers its client proteins to ATP-dependent chaperones for proper refolding, suggesting that mTXNPx acts as chaperone holdase[14]. Expression studies using an *mtxnpx*$^{-/-}$ deletion strain of *L. infantum* confirmed the physiological significance of this chaperone activity[14]. In contrast to mutant parasites expressing wild-type mTXNPx, strains that express a chaperone-*inactive* but peroxidase-*active* variant of mTXNPx were unable to deal with the extensive protein unfolding that they experience when they are forced to adjust to the body temperature of mammals[14]. As a result, these strains failed to propagate in mammalian hosts[14].

To obtain insights into the structural basis and mechanism of mTXNPx chaperone activity, we have now determined the cryo-EM structures of reduced, heat-activated mTXNPx with and without a bound model client protein to 3.7 and 2.9 Å resolution, respectively. We resolved most of the missing N-terminal residues of mTXNPx, and discovered that the N terminus forms a well-resolved central belt surrounding an unstructured and highly flexible client protein in the center of the ring. Our in vivo and in vitro cross-linking studies revealed additional nearby residues that are potentially involved in client binding, and provide new insights into the mechanism by which heat-induced structural rearrangements trigger the activation of mTXNPx chaperone function.

## Results

**Cryo-EM structure of mTXNPx$_{red}$: luciferase complexes**. In our previous work, we used negative-stain EM to visualize complexes formed between reduced mTXNPx decamers and thermally unfolded luciferase[14]. These studies revealed additional density in the center of the decameric ring structure of mTXNPx$_{red}$ in the presence of the client protein but not in its absence. However, the structures lacked the resolution necessary for detailed analysis. Therefore, we sought to determine 3D structures of mTXNPx$_{red}$ by cryo-EM in order to identify the structural basis for mTXNPX$_{red}$-client interactions. Cryo-EM datasets were collected for heat-activated mTXNPx$_{red}$ in the presence or absence of the luciferase model client. 2D class averages revealed well-defined features of the decamer ring and a globular, central density that was present only after incubation with thermally unfolded luciferase (Fig. 1a and Supplementary Figure 1A). We attributed this density to the bound client. This central density has a diameter of ~45 Å and is relatively featureless, which could indicate structural heterogeneity. In contrast, the decameric ring is well defined, revealing five dimeric subunits of mTXNPx that encompass the bound client (Fig. 1a). In side views of the structures solved in the presence of luciferase, the ring appeared slightly wider in the center (arrow, Fig. 1a). This new density did not extend beyond the plane of ring, suggesting that the bound luciferase was either restricted to the region surrounded by the decamer or was too flexible to resolve. Both the architecture and the diameter of the mTXNPx$_{red}$ decameric ring appeared similar with or without bound luciferase, thus indicating that it is unlikely that major conformational rearrangements occur upon client binding. Notably, the dataset for mTXNPx$_{red}$ incubated with luciferase contained a mixture of both luciferase bound and luciferase free mTXNPx$_{red}$ decamers. By comparison of the top-view 2D averages, we determined that ~60% of mTXNPx$_{red}$ was luciferase-bound under these conditions (Supplementary Figure 1B).

We determined 3D reconstructions of mTXNPx$_{red}$ incubated with and without luciferase, which yielded structures at 3.7 and 2.9 Å estimated resolution, respectively (Fig. 1b, c; Supplementary Figure 1C and Table 1). In the final sharpened map, additional central density corresponding to the bound luciferase client was not present at threshold values of >2.0 sigma. However, this density was clearly present in the unsharpened map and in 2D projections of the reconstruction (Fig. 1d and Supplementary Figure 1D). The overall size of the central density attributed to luciferase was consistent with a single 63 kDa luciferase molecule. This conclusion was supported by analytical ultracentrifugation experiments (Supplemental Methods), which showed that one molecule of luciferase is bound per decamer (Supplementary Figure 1E). Angular distribution analysis confirmed that while the top-view orientation of the ring was preferred, additional

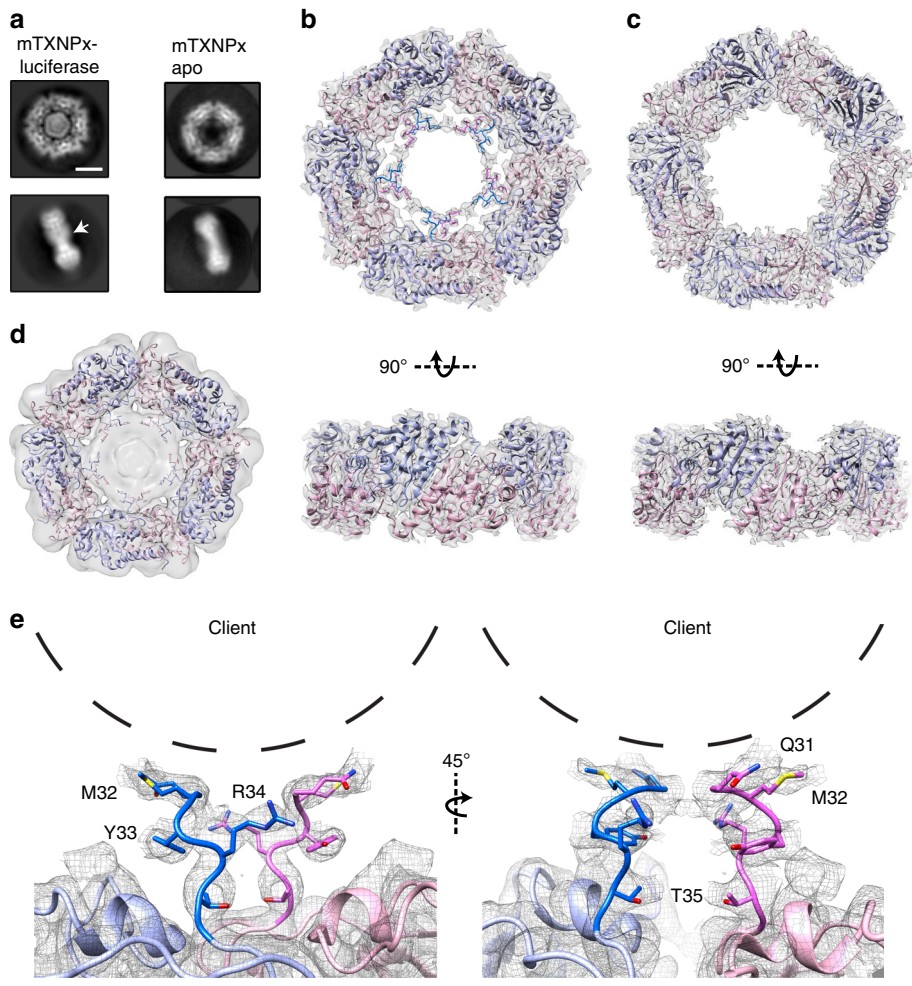

**Fig. 1** Cryo-EM structure of the mTXNPx$_{red}$: luciferase complex. **a** 2D reference-free class averages of heat-treated mTXNPx$_{red}$ in the presence (left) or absence (right) of unfolded luciferase showing additional central density attributed to luciferase in top and side (arrow) views. Scale bar equals 5 nm. Cryo-EM 3D map of the mTXNPx$_{red}$:luciferase complex (**b**) and mTXNPx alone (**c**) resolved to 3.7 and 2.9 Å resolution, respectively shown at $\sigma = 3.3$. Top and side views are shown. The *L. braziliensis* mTXNPx crystal structure (4KB3) was docked into the EM density with additional N-terminal residues, Q31-T35, modeled for mTXNPx$_{red}$:luciferase. **d** The unsharpened map of the mTXNPx$_{red}$:luciferase complex displayed at threshold $\sigma = 1.7$ showing central density corresponding to bound client. **e** N-terminal residues Q31-T35 shown docked into the sharpened mTXNPx$_{red}$:luciferase map with approximate location of client (dashed line) adjacent Q31, M32, Y33, and R34

### Table 1 Summary of atomic modeling statistics and validation scores

| | mTXNPx-apo | mTXNPx-Luciferase |
|---|---|---|
| Amino acid residues[a] | 35–199 | 31–204 |
| RMS deviations | | |
| Bond lengths (Å) | 0.007 | 0.009 |
| Bond angles (°) | 0.954 | 0.984 |
| Ramachandran | | |
| Favored (%) | 94.05 | 92.44 |
| Allowed (%) | 5.95 | 7.56 |
| Outliers (%) | 0 | 0 |
| Molprobity score | 1.72 | 1.85 |
| EMRinger score | 2.55 | 3.23 |

[a]Mitochondrial signal sequence: aa1–27

orientations, including side-views were also present (Supplementary Figure 1F). Local resolution analysis identified uniform higher resolution (~3.6 Å) for the core of the mTXNPx ring and lower resolution regions for the inner face of the ring (i.e., 4.4 Å), indicating structural flexibility in proximity to the bound client

(Supplementary Figure 1G). A homology model from *Leishmania braziliensis* (pdb: 4KB3), a close homolog of *L. infantum*[12], was used for model building and the final model showed good agreement with the cryo-EM density for the core regions of the decamer (Supplementary Figure 1H).

Based on our previous negative-stain EM analysis, we proposed that the N-terminal residues, which are present following cleavage of the mitochondrial targeting sequence (aa 1–27), serve as flexible extensions that likely contact the client in the interior of the decamer ring[14]. However, the first eight residues (aa 28–35) of the N-terminus are not resolved in the dimeric and decameric X-ray crystal structures of *L. braziliensis* Prx1m[12]. Moreover, no other structures of peroxiredoxins have N-termini resolved that extend into the interior of the ring, similar to what we previously observed. Remarkably, in the cryo-EM structure of client-bound mTXNPx$_{red}$, we identified an additional inner ring of density that extends from position T35 and encapsulates the client density (Fig. 1b, d). Residues Q31-T35 of mTXNP were modeled into this density and identified to extend from each peroxiredoxin monomer at the dimer interface to form the inner ring. This inner ring is distinct from the larger decamer structure and consistent with the spoke-like structures that we previously observed[14].

Although we were unable to build a model for this entire region, this inner ring of density is consistent with the eight N-terminal amino acids of the mature peroxiredoxin that were not previously resolved. This N-terminal ring was not observed in the cryo-EM reconstruction of apo mTXNPx$_{red}$ at an equivalent density threshold, indicating that the presence of client likely stabilizes these residues (Fig. 1c).

Based on the position of the N-terminal extensions, residues Y33 and R34 appear to contact the bound client along with additional residues in the N-terminal strand of the chaperone that were unable to be modeled (Fig. 1e). For these reconstructions, D5 symmetry was imposed due to the known symmetry of the mTXNPx$_{red}$ decamer. When refinements were performed without imposing symmetry (C1), the density projections remained present, indicating that all subunits likely make *bona fide* contact with the client, although some variability is observed (Supplementary Figure 1I). In an attempt to better define the client interactions and improve the resolution of the luciferase density, focused refinement was performed using a mask to exclude the outer ring of the decamer (Supplementary Figure 1J). However, density for the bound luciferase did not improve for either symmetrized or unsymmetrized reconstructions. Thus, we conclude that the luciferase client is likely bound heterogeneously and in an unstructured state that, as previously shown, is competent for re-folding[14].

In the mTXNPx$_{red}$ reconstruction in the absence of luciferase, density corresponding to residues R34 and T35 was present as well (Fig. 1b, c). However, we found that these projections were less prominent than in the client-bound complex. The position of this density is identical in both maps, suggesting that these positions do not rearrange, but instead become more ordered upon client binding. In summary, these structures reveal high-resolution views of a chaperone-active peroxiredoxin in complex with a bound client. We found that the N-terminal extensions of mTXNPx$_{red}$ become ordered upon client binding, and directly contact and encapsulate the client. This likely stabilizes the client in an unstructured state that is competent for re-folding in the presence of the DnaK/DnaJ/GrpE system[14].

**mTXNPx's N-terminus is necessary for client binding in vitro**. The cryo-EM structure of mTXNPx$_{red}$ in complex with luciferase suggested that residues in the N-terminal extensions of mTXNPx participate in client binding. To directly determine the extent to which the N-terminal residues of mTXNPx influence its chaperone function, we engineered and purified N-terminal truncation mutants, lacking either the first 5, 8 or 10 residues of the mature protein (Supplementary Figure 2A). We were unable to cleave the N-terminal histidine tag from the Δ8mTXNPx or Δ10mTXNPx proteins but successfully removed the His-tag from the Δ5mTXNPx variant. After cleavage, this mutant variant lacks the five N-terminal aa N$_{28}$LDYQ$_{33}$ from the mature protein but still contains the N-terminal 3-aa scar sequence (i.e., Gly-Ser-His), which is a remnant of the thrombin cleavage and present in all of our purified mTXNPx proteins (Supplementary Figure 2A). In vitro activity assays revealed that this N-terminally truncated mutant variant showed near wild-type like peroxidase activity at 30 °C (Supplementary Figure 2B). In contrast, Δ5mTXNPx$_{red}$ was at least 70% less active as a molecular chaperone in our in vitro aggregation assays as compared to wt-mTXNPx (Fig. 2a). Whereas a 20-fold molar excess of wt mTXNPx$_{red}$ over luciferase completely blocked aggregation, a 40-fold molar excess of the Δ5mTXNPx$_{red}$ variant was necessary to at least partially suppressed luciferase aggregation. These results agreed well with our cryo-EM data and further illustrated the importance of mTXNPx's N-terminal extensions in chaperone function.

Two other residues, i.e., Y33 and R34, also appeared to be involved in client interactions according to our cryo-EM structure but were not covered by our 5-aa truncation. To directly test their role in in vitro client binding, we substituted the respective residues individually with Ala, purified the two mutant variants (mTXNPxY33A, mTXNPxR34A) as before, and tested for their chaperone activity. We found that while replacing the bulky hydrophobic tyrosine with the smaller alanine slightly decreased the chaperone activity of mTXNPx (Fig. 2b, left panel), substituting the charged R34 with Ala significantly increased the ability of mTXNPx to prevent luciferase aggregation (Fig. 2b, right panel). These mutations only mildly (Y33A) or not significantly (R34A) altered the peroxidase activity at 30 °C (Supplementary Figure 2B). Finally, we tested the chaperone activity of the two *L. infantum* cytosolic peroxiredoxins isoforms, which are highly homologous to the mitochondrial protein but lack the N-terminal 4 aa sequence "NLDY" and harbor "CG" instead of "YR" at positions 33/34 (Supplementary Figure 2C). As shown in Supplementary Figure 2D, neither one of the two cytosolic homologs showed any detectable activity in the in vitro chaperone assay. In summary, these results revealed that truncating the N-terminal extensions and/or altering the hydrophobicity of residues facing inside of the decameric ring, greatly affects client binding, supporting the conclusion that the flexible N-termini play a crucial role in the chaperone function of *Leishmania* mTXNPx.

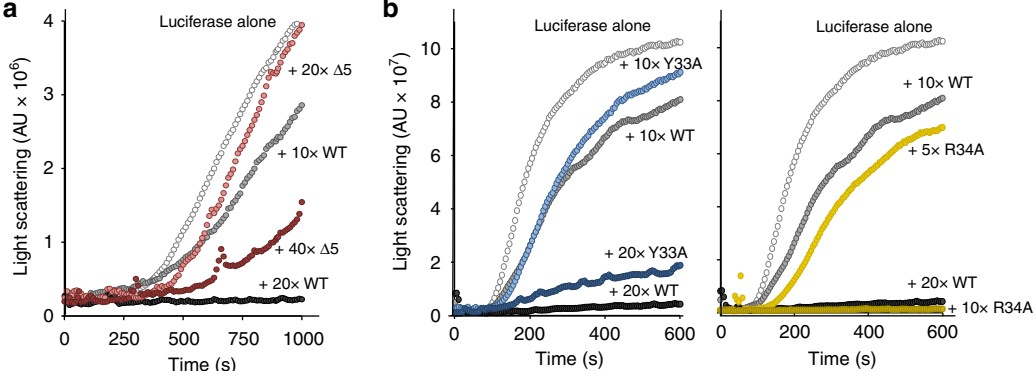

**Fig. 2** N-terminal extensions of mTXNPx are crucial for in vitro chaperone activity. Influence of reduced Δ5mTXNPx (**a**) or mTXNPxY33A/mTXNPxR34A (**b**) on the thermal aggregation of luciferase. Native luciferase (0.1 μM) was incubated in the absence or presence of different ratios of mTXNPx at 42 °C. Light scattering (expressed in arbitrary units, A.U.) was monitored at 360 nm. All reactions were carried out in the presence of 0.2 mM DTT to maintain reduction of mTXNPx. Purified wt mTXNPx was used as positive control

**Incorporation of an unnatural amino acid for in vivo cross-linking**. To independently evaluate mTXNPx$_{red}$-client-interaction sites in the cellular context, and test whether in vivo clients engage in interactions that involve residues within and/or beyond the inner ring structure, we utilized an in vivo cross-linking technique that was originally developed by Schultz et al.[20,21]. By applying site-specific mutagenesis to mTXNPx, we individually substituted each phenylalanine or tyrosine codon in the mTXNPx with a TAG amber stop codon. Upon co-transformation of the mutant plasmids together with the aminoacyl-tRNA synthetase/suppressor tRNA pair into E. coli, and addition of the unnatural, UV-cross-linkable amino acid p-benzoyl-L-phenylalanine (Bpa) to the growth media, these mutant variants incorporate Bpa and hence contain potential cross-linking sites at 19 distinct sites that are distributed throughout the amino acid sequence of mTXNPx (Fig. 3a, Supplementary Figure 3A). We reasoned that mTXNPx would, in general, be likely to tolerate these substitutions since Bpa is a structural analogue of Phe/Tyr. Once activated by UV, Bpa acts as zero-length cross-linker, which interacts irreversibly with other residues but only if they are in very close proximity to each other.

Upon co-transformation of the plasmids, we tested for in vivo Bpa incorporation into mTXNPx by growing the cells in the presence or absence of exogenously-added Bpa at 30 °C for 3 h. Next, we analyzed the expression level of full-length mTXNPx by western blot analysis using anti-mTXNPx antibodies. We found that the majority of our E. coli strains expressed some measurable amounts of full-length mTXNPx in the presence of Bpa but not in its absence (Supplementary Figure 3B). The expression levels of Y63Bpa, F192Bpa, and F194Bpa were reproducibly low. The F221Bpa variant, whose Bpa substitution is located only 5 aa from the C-terminus of mTXNPx was also detectable in the absence of Bpa, making it impossible to determine whether this variant, and by extension the F222Bpa variant, contained Bpa or not.

In vitro studies have shown that mTXNPx is only activatable as chaperone when present in the reduced form[14]. To assess the in vivo oxidation status of mTXNPx when expressed in the cytosol of E. coli under both non-stress and stress conditions, we conducted in vivo thiol-trapping experiments using N-ethylmaleimide (NEM) followed by SDS-PAGE of the extracts under non-reducing conditions[4]. This approach was based on previous results that revealed that under these conditions, reduced mTXNPx migrates in the monomeric form while oxidized mTXNPx migrates as a disulfide-linked dimer[22]. As shown in Supplementary Figure 3C, the majority of NEM-labeled mTXNPx migrated in its monomeric (i.e., reduced) form, and no change in its oxidation status was observed upon exposure to 45 °C. We obtained very similar results with our NEM-labeled mTXNPx-Bpa variants, indicating that they were present in their presumably chaperone-activatable reduced conformation in vivo as well (Supplementary Figure 3D). Moreover, we did not observe any higher molecular-weight complexes that would indicate the formation of disulfide-linked mTXNPx-client complexes. These results argue against the possibility that mTXNPx undergoes significant thiol-disulfide exchange reactions with other thiol-containing proteins in E. coli.

To ascertain that the introduction of Bpa at distinct sites in the protein does not affect the chaperone function of mTXNPx, we purified two Bpa-mutants (i.e., Y73Bpa, Y111Bpa) and investigated their effects on the thermal aggregation of luciferase (Supplementary Methods). As shown in Supplementary Figure 3E, both Bpa mutant variants behaved like wild-type mTXNPx in the chaperone assay. This result, the fact that Bpa is structurally very similar to Phe and Tyr, and the finding that all Bpa-variants are expressed in a soluble form in vivo, gave us confidence that incorporation of the unnatural amino acid Bpa did not profoundly affect the function of mTXNPx.

**mTXNPx residues involved in client binding in vivo**. Previous in vitro studies have revealed that reduced, decameric mTXNPx becomes rapidly activated upon exposure to heat shock temperatures, and forms apparently stable complexes with thermally unfolding client proteins[14]. We reasoned that shift of our mTXNPx-Bpa expressing E. coli strains from 30 to 45 °C should activate reduced mTXNPx and, at the same time, generate a pool of thermally unfolding client proteins that could potentially interact with active mTXNPx. To test for in vivo cross-linking, we cultivated all of our mTXNPx-Bpa expressing strains as before in Bpa, and either left the cells at 30 °C or shifted them to 45 °C for 30 min. After the incubation, we moved the samples onto ice. We did this because of the prior observation that in vitro complexes between mTXNPx and client proteins are stable over prolonged time[14]. To cross-link potential in vivo clients with mTXNPx, we then subjected the samples to a 10-min exposure to UV light, applied the samples onto reducing SDS Page gels and detected mTXNPx by western blotting (Fig. 3b). The most prominent band that we detected upon UV-cross-linking in all but one mutant Bpa variant (i.e., Y63Bpa) was independent of temperature, and migrated at a position corresponding to that of twice the molecular weight of a mTXNPx monomer, suggesting that it corresponds to a Bpa-linked mTXNPx dimer. In addition, however, some lysates revealed higher molecular-weight species, which became particularly pronounced when the bacteria were shifted to 45 °C. We reasoned that—based on the architecture of mTXNPx—cross-links within mTXNPx should only be possible across either the A-type or the B-type interface but should yield no inter-mTXNPx cross-links that are higher in molecular weight than dimers. We therefore deduced that the smear of higher migrating mTXNPx-positive bands that we observed in select mTXNPx-Bpa expressing strains most likely represent cross-linked complexes between select mTXNPx-Bpa variants and in vivo client proteins (mTXNPx-X) (Fig. 3b).

A comparison of the cross-linking patterns and efficiencies of different mTXNPx mutant variants at 30 °C and 45 °C revealed that the Bpa-variants fit into three different categories (Fig. 3b; summary of all cross-linking results can be found in Supplementary Table 1). Group 1 Bpa variants (F45, Y67, F71, F72, Y73, F77, F79, F88, and F95) reproducibly showed a significant increase in the amount of higher-migrating bands upon incubation at 45 °C as compared to what was found at 30 °C (Fig. 3, indicated by red bars). This suggests that these residues are the most important for temperature-dependent in vivo client binding. Group 2 Bpa variants formed irreversible cross-links with other proteins upon UV-exposure independent of the incubation temperature (Fig. 3b, indicated by magenta bars). Surprisingly, the only variants that behaved in this manner were those that contained Bpa substitutions in the two N-terminal tyrosine residues (Y30, Y33), which contact thermally unfolding luciferase in vitro according to our cryo-EM model. Group 3 Bpa variants (Y63, Y111, Y145, F160, and F192) did not reproducibly show any higher migrating smear or formed only a small number of discrete higher-migrating bands (Fig. 3b, indicated by blue bar), suggesting that these residues do not usually come in direct contact with in vivo clients. One Bpa-variant (Y194) showed inconclusive cross-linking results (Supplementary Table 1) and was disregarded for further analysis.

**Dimer–dimer interface serves as transient interaction site**. To visualize which region(s) of mTXNPx in addition to the N-terminus might be involved in in vivo client binding, we mapped

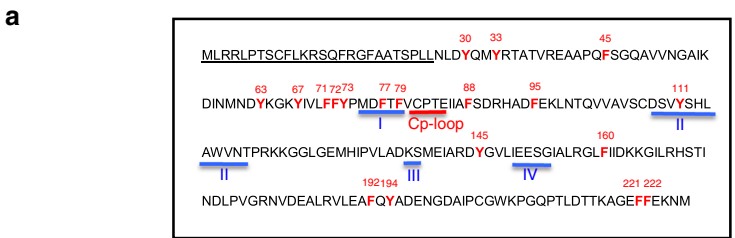

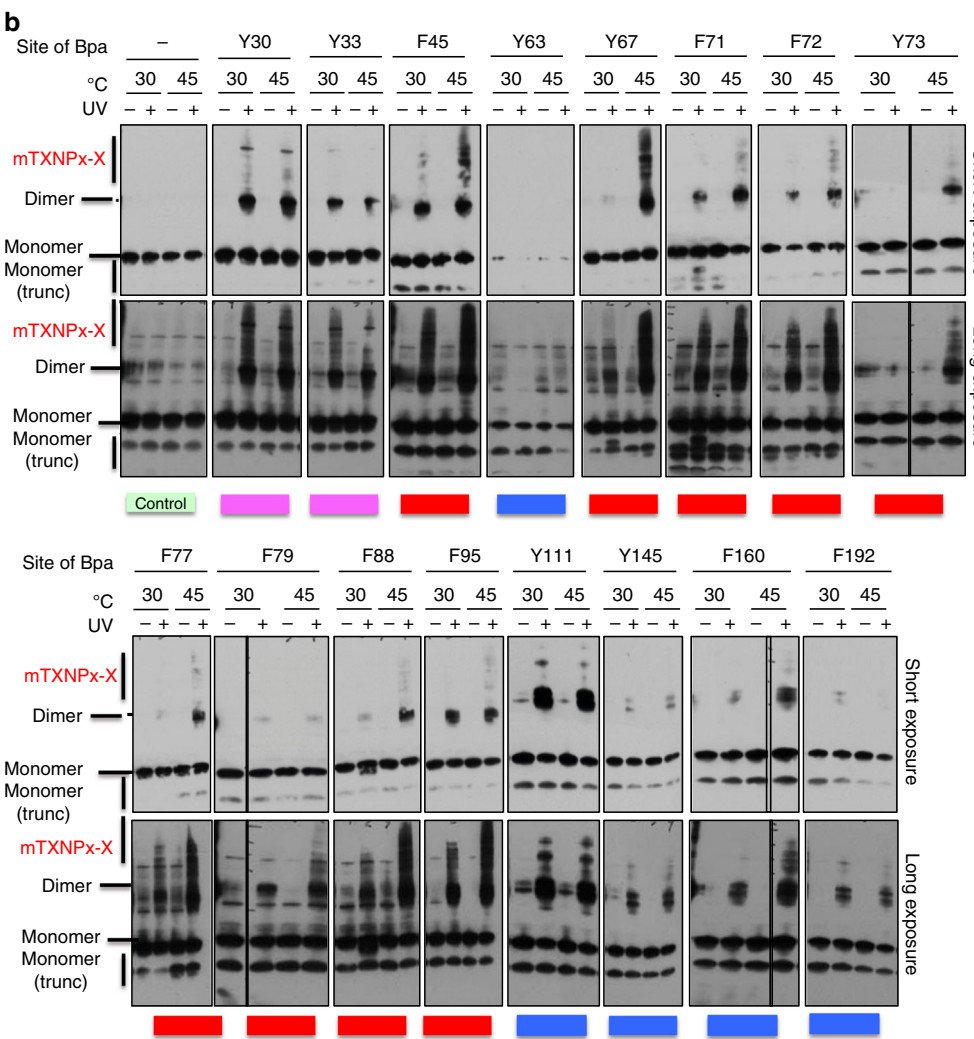

**Fig. 3** UV-cross-linking of mTXNPx to client proteins in vivo. **a** Bpa substitutions in mTXNPx. Phe and Tyr residues (indicated in red) were individually mutated to allow for the site-specific incorporation of the photo-activatable amino acid Bpa. The $C_p$-loop-helix, which contains the peroxidative Cys81 and which is folded when mTXNPx is reduced and decameric, is underlined in red. All regions (regions I–IV) involved in dimer–dimer interactions are underlined in blue. The mitochondrial targeting sequence is underlined in black. The targeting sequence is not present in the in vivo constructs. **b** In vivo cross-linking of mTXNPx-Bpa variants. *E. coli* expressing either wild-type mTXNPx or the mTXNPx-Bpa variants were grown at 30 °C and either kept at 30 °C or shifted to 45 °C for 30 min. Afterwards, aliquots of the cells were either left untreated (−) or were exposed (+) to UV light for 10 min to induce cross-linking. Subsequently, bacteria were lysed and cell extracts were analyzed by western blot using anti-mTXNPx antibodies. Two different exposure times are shown. The two most prominent bands corresponded to monomeric (mTXNPx) and dimeric mTXNPx-mTXNPx, respectively. Additional bands, seen as higher-migrating smear, are suggestive of mTXNPx-*E. coli* protein cross-links (mTXNPx-X). A representative experiment is shown (results from all experiments conducted are summarized in Supplementary Table 1). Based on the extent of higher migrating smear (i.e., cross-links) upon incubation at 30 °C and/or 45 °C, the residues were classified into three groups: Group 1: Bpa-variants that cross-link more extensively upon incubation at 45 °C compared to 30 °C (red); Group 2: Bpa-variants that show equal cross-linking at 45 °C and 30 °C (magenta); Group 3; Bpa-variants that either do not show any reproducible higher migrating smear, or show only a small number of discrete higher migrating bands (blue). Bpa-194, Bpa-221, and Bpa-222 variants either produced inconsistent results or were not reliably produced in vivo

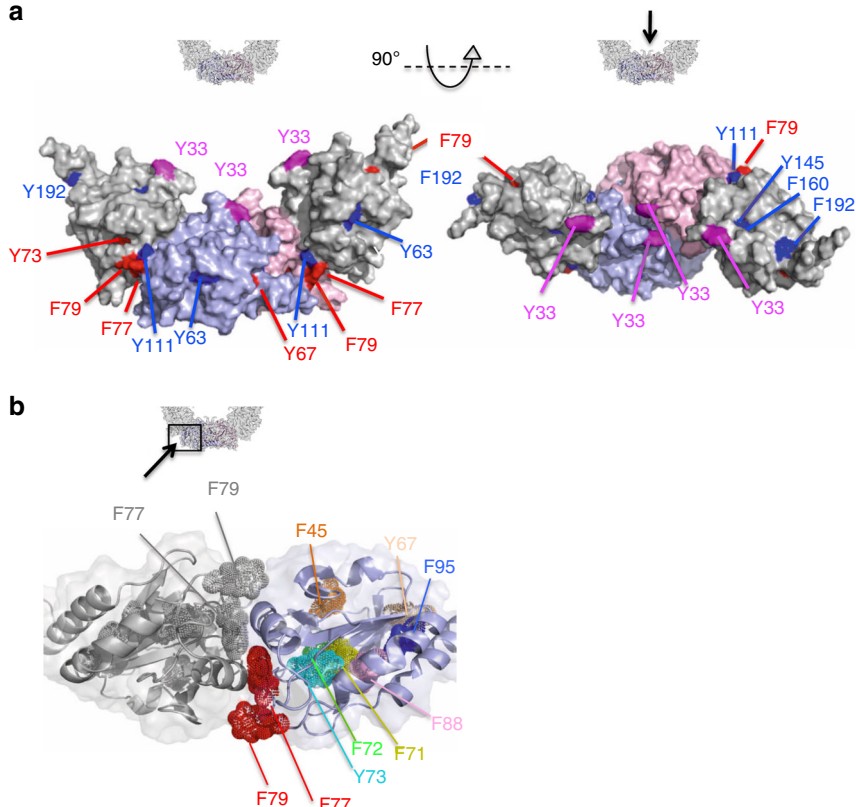

**Fig. 4** Mapping of in vivo cross-linking sites onto quaternary structure of mTXNPx. **a** Quaternary structure of reduced *L. infantum* mTXNPx$_{red}$. For simplicity, only one of the five dimers that are present in the decamer (subunit A in pink and subunit B in blue) as well as one adjacent subunit on each side (grey) are shown. Different views of the selected mTXNPx dimer are presented. Amino acids that, when replaced by Bpa, reproducibly produce higher migrating smear upon UV-cross-linking either after shift to 45 °C (Group 1, red), at both 30 °C and 45 °C (Group 2, magenta) or no shift (Group 3, blue) are shown. **b** Close-up view of the dimer-dimer interface and location of Group 1 residues. Group 1 residues F77 and F79 (red and grey dots) form part of the dimer-dimer interface. All other group 1 residues (in rainbow colors) are mostly buried

our Bpa variants onto the solved cryo-EM structure of *L. infantum* mTXNPx (Fig. 4a). For simplification, we only show one dimeric unit of the decamer (monomers shaded in light blue or pink, respectively) encompassing the B-type monomer–monomer interface, and one neighboring subunit on each side (shaded in grey) to illustrate contacts in the A-type dimer–dimer interfaces[12]. We applied the same color scheme we had used in Fig. 3. As expected, the two Group 2 residues, i.e., Y30, Y33 face the interior of the decameric ring (Fig. 4a, magenta) while most of the non-cross-linking Group 3 residues were buried. Three of them (i.e., Y145, Y160, and Y192) clustered in the monomer–monomer interface, explaining their ability to crosslink mTXNPx dimers but not client proteins (Fig. 4a, blue). To our surprise, however, most of our Group 1 residues, which formed higher molecular weight cross-links specifically upon shift to elevated temperatures, were also not surface accessible according to our cryo-EM structure (Fig. 4a, red; Supplementary Table 1). In fact, six of these residues form an almost contiguous hydrophobic region (Fig. 4b, rainbow colors) that is masked by several loops, which appear to be partially stabilized by dimer–dimer contacts. The other two Group 1 residues (i.e., F77, F79) are located in the highly conserved C$_p$-loop-helix active site motif of region 1[23], which forms a surface complimentary to the interfacial region 2 of the opposing dimer (Fig. 4b, red spheres). Residue F77 is deeply buried within this largely hydrophobic 600 Å interface that is known to play a major role in dimer–dimer stabilization. These results raised the intriguing question as to how these residues, which are buried either directly within the dimer–dimer interface

or by structural elements that are stabilized by dimer–dimer contacts, become accessible for cross-linking to client proteins during activating conditions in vivo. Previous studies provided evidence that unfolding of the C$_p$-loop-helix, which is typically favored by oxidation processes, low pH or phosphorylation events, leads to the rearrangement of the decamer, including its dissociation into dimers[7,9,23]. Based on these findings and our results, we now hypothesized that temperature-induced activation of decameric mTXNPx$_{red}$ might destabilize the hydrophobic dimer–dimer A-type interface, thereby exposing binding sites for unfolding proteins. This mechanism is reminiscent of small heat shock proteins (sHsps), which also undergo stress-activated dissociation followed by client-mediated re-association events[24,25].

**In vitro cross-linking experiments support in vivo results.** To independently validate these additional interaction sites and to potentially gain more insights into conformational rearrangements that mTXNPx$_{red}$ undergoes upon heat-induced activation and/or client binding, we conducted in vitro cross-linking experiments. We used two different primary amine-reactive cross-linkers, DiSuccinimidylSubarate (DSS) and DiSuccinimidylAdipate (DSA) followed by a tryptic digestion and MS/MS analysis. We incubated decameric mTXNPx$_{red}$ with thermally unfolded luciferase at 42 °C as before, separated insoluble proteins by centrifugation, and cross-linked the soluble proteins using either DSS or DSA (see Supplementary Figure 4A for cross-linking scheme). As a control, we used oxidized dimeric

mTXNPx$_{ox}$, which neither functions as a heat-shock activated chaperone nor undergoes conformational rearrangements upon incubation at heat shock temperatures in vitro[14]. As expected, we found that only the presence of mTXNPx$_{red}$ was able to protect luciferase against thermal aggregation and to maintain the protein in its soluble form (Supplementary Figure 4A, lane 9). Upon cross-linking with DSS or DSA, we observed specific higher molecular weight complexes, suggestive of mTXNPx$_{red}$–luciferase complexes (Supplementary Figure 4A). We excised and extracted these higher molecular-weight complexes from the gel, digested the proteins with trypsin and analyzed the peptides by MS/MS analysis (Supplementary Data 1). The experiments were designed so that data sets contain information about *intra*-protein cross-links, i.e. cross-links within each of the two proteins, as well as *inter*-protein cross-links that connect amine-containing residues in mTXNPx$_{red}$ with those of luciferase.

Analysis of *intra*-protein cross-links revealed the expected cross-links between physically close lysine residues in either mTXNPx$_{red}$ or luciferase (Fig. 5a, Supplementary Figure 4B, purple lines; Supplementary Data 1). In luciferase, we also detected a number of cross-links that could not be explained by proximity (Supplementary Figure 4B, right panel). These longer-range cross-links (>20 residues apart in the primary sequence) support the notion that luciferase is in a partially unfolded state when bound to mTXNPx$_{red}$. Analysis of the *inter*-protein cross-links between mTXNPx$_{red}$ and luciferase revealed that the vast majority of cross-links involves residues in the N-terminus of mTXNPx$_{red}$ and different regions in luciferase (Fig. 5a, cyan lines; Supplementary Data 1). These findings were in excellent agreement with our previous results. In addition, however, we also identified three residues to cross-link with luciferase that were not located in the N-terminus. Although their respective Kojak scores were below our arbitrary cut-off of 2, the fact that we detected 10 different peptides involving K165, seven different peptides involving K123 and three different peptides involving K137 makes us reasonably confident that these residues in mTXNPx are actually involved in cross-linking with luciferase as well. All three residues are found in close proximity to our previously identified in vivo interaction sites (Fig. 5b, indicated in orange). While both K123 and K137 are located immediately adjacent to F77/F79 in the A-type interface, K165 localizes near Y67, another Group 1 residue (Fig. 5b). These results support our conclusion that client interactions not only engage the N-terminal extensions of mTXNPx$_{red}$ but also involve other regions, including residues close to or part of the dimer–dimer interface.

**Elevated temperatures destabilize the dimer–dimer interface.** Both in vivo and in vitro cross-linking experiments implicated the dimer–dimer interface of mTXNPx$_{red}$ in client interactions. We therefore wondered whether temperature-induced activation of decameric mTXNPx$_{red}$ could lead to the destabilization of the dimer–dimer interface, causing potentially-buried hydrophobic regions to turn into (transient) binding sites for unfolding proteins. Upon closer inspection of the decameric structure of mTXNPx$_{red}$, we identified two lysine residues (K122/K123), which are positioned in very close proximity (12.1–12.9 Å) to the two corresponding lysine residues of the neighboring dimer (Supplementary Figure 5A). We now reasoned that any significant rearrangement of the decamer might alter the extent to which these lysine residues become cross-linked by amine-reactive cross-linker (CL). We therefore decided to conduct quantitative cross-linking (qCL) experiments using the isotopically coded forms of either DSS (DSS-$^{12}C_6$/$^{13}C_6$) or DSA (DSA-$^{12}C_6$/$^{13}C_6$). As a proof-of-concept experiment, we first cross-linked dimeric mTXNPx$_{ox}$ with heavy-labeled CL and

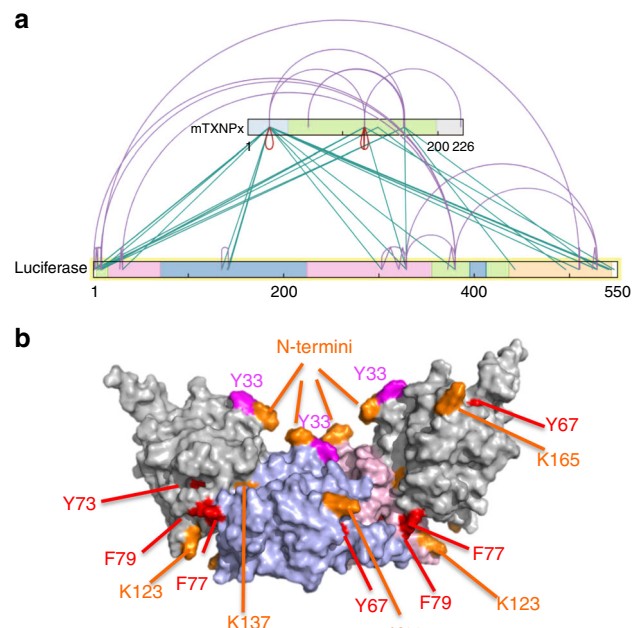

**Fig. 5** In vitro cross-linking of mTXNPx$_{red}$-luciferase complexes. **a** *Intra*-protein (purple lines) and *inter*-protein (cyan lines) cross-link maps using xiNET (Supplementary Data 1). The mTXNPx sequence including the mitochondrial targeting sequence was used for the map. **b** Mapping of the in vitro cross-linking sites between mTXNPx$_{red}$ and luciferase (shown in orange) and the in vivo cross-linking sites between mTXNPx-Bpa variants and cellular client proteins (red: Group 1 cross-links; pink: Group 2 cross-links, Fig. 4a) onto the quaternary structure of *L. infantum* TXNPx$_{red}$. For simplicity, only one of the five dimers that are present in the decamer (subunit A in pink and subunit B in blue) as well as one adjacent subunit on each side (grey) are shown

decameric mTXNPx$_{red}$ with light-labeled CL at 30 °C, and mixed the samples 1:1. After tryptic digest and MS/MS analysis, we then calculated the mass spectrum signal intensity ratios of the heavy to light forms (i.e., H/L-ratios) focusing on inter-protein cross-linked peptide pairs containing the lysines of interest, i.e., K122/K123. By using DSS as CL, we identified two different inter-protein K123-K123 peptide pairs, which both revealed negative log2-ratios (i.e., −2.09 and −1.71) (Table 2). These results indicated that the light-labeled cross-link (i.e., the decameric form) is favored (a complete list of all qCL results can be found in Supplementary Data 2). Very similar results were obtained when we conducted the same experiments with the isotopically coded forms of DSA (Supplementary Table 2). These results confirmed that inter-protein cross-links between K123 residues of neighboring dimers are favored when mTXNPx is decameric and disfavored when mTXNPx is dimeric. To now test how shifting reduced mTXNPx decamers from 30 to 42 °C affects the cross-linking behavior of these neighboring residues, we incubated mTXNPx$_{red}$ in the presence of either the isotopically heavy or the light form of the cross-linker at 30 °C or 42 °C and mixed the samples according to the flow chart outlined in Supplementary Figure 5B. In one set-up, for instance, we combined mTXNPx$_{red}$ that was incubated at 42 °C and cross-linked with heavy DSS with a sample containing mTXNPx$_{red}$ that was incubated at 30 °C and cross-linked with light DSS (Table 2). As before, we quantified the relative changes in inter-protein cross-links after tryptic digest and mass spectrometric analysis. We identified three different K123-K122/K123 inter-protein peptides, all of which gave negative H/L log2 ratios (−1.55 to −2.57), indicating that the light-labeled cross-links (i.e., 30 °C incubation) are favored (Table 2).

**Table 2 Quantitative cross-linking (qCL) of mTXNPx$_{red}$ and mTXNPx$_{ox}$ using DSS**

| Light-CL | Heavy-CL | CL-Peptides | Log2 (H/L)-average | Favored CL |
|---|---|---|---|---|
| Red (30 °C) | Red (30 °C) | 1–1 | −0.04 | – |
| Red (30 °C) | Red (30 °C) | 1–1 | −0.02 | – |
| Red (30 °C) | Ox (30 °C) | 1–3 | −2.09 | Red (30 °C) |
| Red (30 °C) | Ox (30 °C) | 1–1 | −1.71 | Red (30 °C) |
| Red (42 °C) | Red (30 °C) | 1–2 | 1.40 | Red (30 °C) |
| Red (30 °C) | Red (42 °C) | 1–3 | −2.57 | Red (30 °C) |
| Red (30 °C) | Red (42 °C) | 1–1 | −2.32 | Red (30 °C) |
| Red (30 °C) | Red (42 °C) | 1–3 | −2.21 | Red (30 °C) |
| Red (30 °C) | Red (42 °C) | 1–2 | −1.55 | Red (30 °C) |

*Pep1* K$_{123}$GGLGEMHIPVLADK, *Pep 2* K$_{122}$K$_{123}$GGLGEMHIPVLADK, *Pep 3* K$_{123}$GGLGEM*HIPVLADK, * +16 Da

These results strongly suggested that incubation of decameric mTXNPx$_{red}$ at heat shock temperatures weakens the dimer–dimer interface and may potentially lead to the dissociation of mTXNPx$_{red}$ into reduced dimers. Again, we obtained very similar results with DSA as crosslinker (Supplementary Table 2) and equivalent results when we tested other combinations (Supplementary Figure 5B, Supplementary Data 2). It is of note that in addition to the observed changes in inter-protein cross-links, we also found at least two intra-protein cross-linked peptides that were specifically underrepresented in heat-treated mTXNPx$_{red}$ compared to mTXNPx$_{red}$ that was incubated and cross-linked at 30 °C (Supplementary Data 2). These results suggested that structural changes occur also within the mTXNPx-dimers upon heat-induced activation that lead to altered cross-linking behavior.

To directly test whether mTXNPx$_{red}$ decamers dissociate at heat shock temperatures, we conducted in vitro cross-linking experiments with glutaraldehyde (GA)[26]. We determined cross-linking conditions at 30 °C under which the majority of mTXNPx$_{red}$ molecules cross-linked in the decameric form while the remainder cross-linked in the dimeric form (Fig. 6a). When we conducted the same incubation and cross-linking at 42 °C, the overall cross-linking efficiency did not change. However, upon shift to 42 °C, less then 15% of mTXNPx$_{red}$ molecules cross-linked as decamers while the remaining 85% of molecules cross-linked as dimers (Fig. 6a, b). Subsequent cooling of the sample from 42 to 4 °C followed by cross-linking at 30 °C re-established the decameric ross-link, indicating that the dissociation is reversible (Fig. 6a, b). Presence of luciferase increased the cross-linking yield of the decamers upon shift to 42 °C, suggesting that client-binding stabilizes the decamer (Fig. 6a, b). We obtained very similar results when we applied mTXNPx$_{red}$ or mTXNPx$_{red}$-luciferase complexes to negative-stain EM grids immediately after incubating the proteins at elevated temperatures. While ~50% of the mTXNPx$_{red}$-decamers dissociated upon heat treatment, cooling down the samples before applying them onto the grid restored the decamer population (Fig. 6c, d). Moreover, presence of client proteins revealed stabilizing effects under these experimental conditions as well (Fig. 6c, d). These results provide strong evidence that mTXNPx$_{red}$ undergoes reversible temperature-induced conformational changes that are likely to be directly involved in the chaperone activation mechanism of mTXNPx$_{red}$.

## Discussion

Although numerous studies have provided evidence that certain 2-Cys Prx convert from peroxidases to chaperones during oxidative, heat or low pH stress[8,19,27], very little information exists regarding the mechanism(s) by which they do so. In addition, it is unclear what regions in those 2-Cys Prx serve as chaperone-client binding sites. We previously proposed that the reduced, decameric pool of mitochondrial 2-Cys Prx mTXNPx, which turns into a potent chaperone upon incubation at heat shock temperatures, uses residues facing the inside of the decameric ring as binding sites for chaperone clients[14]. Two pieces of data supported this conclusion: first, negative-stain electron microscopy images of mTXNPx$_{red}$ and pre-formed mTXNPx$_{red}$-client complexes revealed client-dependent density in the center of the decameric ring[14]. Second, we found that addition of a 6-His-tag to mTXNPx's N-terminus, which has been proposed to face the interior of the ring structure, abrogates the in vitro and in vivo chaperone activity of mTXNPx but not its peroxidase function[14]. In fact, we used this chaperone-inactive mTXNPx mutant variant to demonstrate that the chaperone activity is the crucial function that allows survival of the parasites inside warm-blooded mammalian hosts[14]. Limitations of the previous studies, however, included the lack of structural information regarding the position and involvement of the N-termini in parasitic Prx, due to their intrinsic flexibility, and the fact that no additional structural information was available to directly tie the N-termini to client interactions. In the current study, therefore, we used a range of independent techniques, including cryo-EM structure analysis of mTXNPx-client complexes, mutational studies as well as in vivo and in vitro cross-linking, to identify client-interaction sites and gain insights into the structural rearrangements that trigger the switch from peroxidase to chaperone function in mTXNPx.

We have now obtained what, to our knowledge, represents the first high-resolution cryo-EM structure of the mTXNPx chaperone-client complex. Our approach has not only allowed us to solve the structure of much of the previously missing N-terminus of mTXNPx but revealed that the N-terminal residues form a belt-like structure, which apparently stabilizes the unfolded client protein. Our mutational analysis and cross-linking studies supported these results, identifying the N-terminal residues of mTXNPx to be important for in vitro chaperone activity, as well as for coming into sufficiently close contact with both in vivo and in vitro client proteins to promote site-specific cross-linking. Similar flexible extensions have been found in several other chaperones, particularly the sHsps, where they are also thought to play a dominant role in client recognition and/or binding[28,29].

Current chaperone models suggest that flexible, intrinsically disordered extensions mediate the initial, potentially low-affinity and long-range interactions with client proteins, and that these interactions are then enforced and strengthened by hydrophobic interactions[30,31]. Our observation that the two N-terminal residues Y30 and Y33 cross-link with potential in vivo clients even at 30 °C supports the idea of the N-termini playing a role in low-affinity interactions. In addition, we identified several other residues that cross-linked with client proteins. To our surprise,

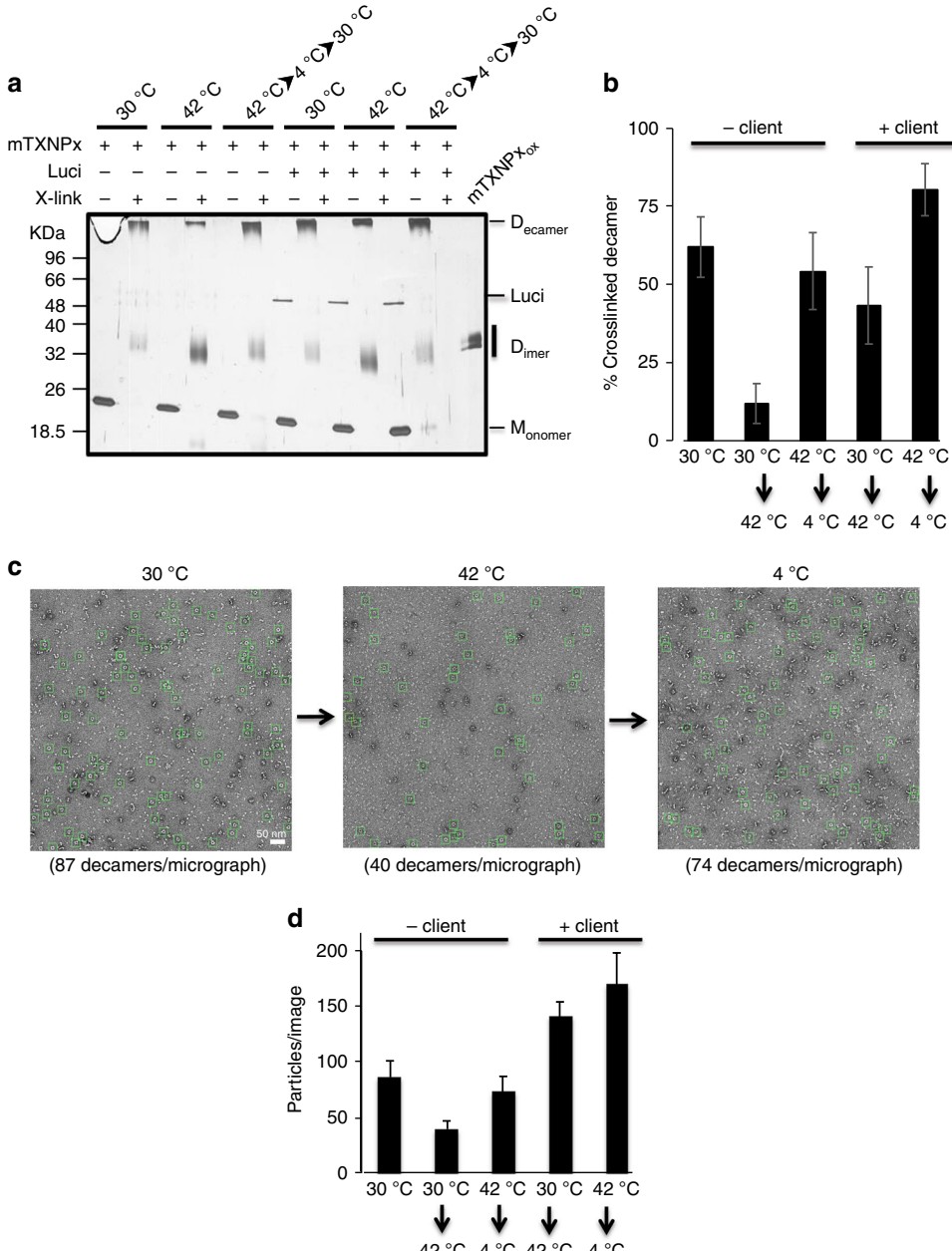

**Fig. 6** mTXNPx decamers undergo heat-shock-induced rearrangements. **a** mTXNPx$_{red}$ (2 μM) either alone or in complex with luciferase (0.2 μM) was incubated in the presence (+) or absence (−) of 40 mM glutaraldehyde at 30 °C or 42 °C for 10 min. The oligomeric status of mTXNPx$_{red}$ was assessed by running the samples on non-reducing SDS-PAGE followed by silver staining. Dimeric mTXNPx$_{ox}$ was used as control. In the absence of cross-linker, mTXNPx$_{red}$ migrates in the monomeric form. **b** Three independent experiments were quantified using ImageJ software. Values represent means ± SD. **c** mTXNPx$_{red}$ (10 μM) was incubated either alone or in the presence of luciferase (1 μM) for 10 min at 42 °C. One aliquot was immediately applied onto an EM grid while the other aliquot was cooled down before applying onto the grids. **d** Quantification of the number of decamers present in 10 micrographs for each condition +/− std is shown

however, according to our cryo-EM structure, most of these residues appeared to be inaccessible in the substrate-free structure of mTXNPx, either shielded by loop regions or directly involved in the hydrophobic dimer–dimer interface that stabilizes the decamer. These results implied that activation of mTXNPx's chaperone function involves substantial conformational rearrangements that at least transiently expose buried hydrophobic sites for binding to client proteins. This conclusion was consistent with our previous negative-stain EM studies, where side-view projections of mTXNPx-luciferase complexes revealed client-associated density specifically protruding from only one side of the decameric ring[14]. Similarly, our cross-linking studies revealed cross-links between mTXNPx and client proteins that specifically engage residues at only one of the two mTXNPx surfaces that are perpendicular to the central ring opening (Fig. 4). At this point, we do not know why our cryo-EM approach was not capable of capturing these additional mTXNPx-client interactions, or why we were unable to observe any major conformational rearrangements in the client-bound mTXNPx that were predicted based on previous structural analyzes[14].

One region in mTXNPx that reproducibly cross-linked with clients both in vivo or in vitro involved residues in the highly

conserved active site $C_p$-loop-helix, whose folding state has been previously shown to play a major role in regulating the oligomeric state and functional activity of Prx. The $C_p$-loop-helix, which is folded in reduced, decameric Prx, has been shown to locally unfold not only upon oxidation[5,7] but also in response to either low pH[9], overoxidation of the peroxidatic cysteine Cp[8,9], or in response to phosphorylation[13,32]. Each of these events (apart from oxidation) has also been shown to trigger the conversion of 2-Cys-Prx into the chaperone-active form. Partial unfolding of the $C_p$-loop-helix in response to oxidation disrupts the dimer–dimer interface, thereby promoting the dissociation of 2-Cys-Prx into the respective dimers. Although stress-induced unfolding of the $C_p$-loop-helix was shown to lead to the formation of higher oligomeric states (i.e., spherical arrays of decamers, filaments), it is entirely possible that it too triggers initial dissociation into dimers, which subsequently re-associate into higher oligomeric states. We now propose that mTXNPx might sense elevated temperatures directly through the folding status of the $C_p$-helix, which would trigger rearrangements in the dimer–dimer interface and could potentially explain how otherwise buried residues become accessible for client interactions. Our in vitro qCL and negative-stain EM experiments support this model, and showed that client-free mTXNPx dissociates into reduced dimers upon incubation at elevated temperatures. The presence of client proteins either prevented the dissociation or caused rapid re-association of the dimers. Given that the stability of the $C_p$-loop-helix together with specific structural features in the C-terminus appear to define which members of the 2-Cys-Prx family are sensitive towards hyperoxidation and which ones are not (for a recent review see ref. [32]), we hypothesize that some members of the 2-Cys-Prx families, such as mTXNPx, have evolved features to directly use the folding status of this helix to sense elevated temperatures. It remains to be demonstrated if destabilization of the $C_p$-helix in mTXNPx or other 2-Cys-Prx is sufficient to turn these proteins into chaperones without the need for any other cues.

Two-Cys-Prxs such as mTXNPx share many intriguing features with members of the sHsp family[24]—like 2-Cys-Prx, sHsps form high molecular-weight oligomers of various sizes, use dimers as their basic structural units, and undergo stress (i.e., heat)-induced dissociation into their dimeric subunits (e.g., Hsp26)[24,25]. We now postulate that, like sHSPs, mTXNPx$_{red}$ is a chaperone that is directly regulated at the protein level by heat shock. Although dissociation into dimers appears not to be mandatory for sHsps chaperone function[33,34], it appears to contribute to increased chaperone activity in some members of the family[24,25]. Once associated with clients, both mTXNPx and sHsps are found re-assembled into higher molecular weight complexes, binding typically one client protein per oligomer[24]. Both proteins have been shown to use flexible N-terminal extensions to engage in long-range interactions with unfolding client proteins[28,29,35] as well as use regions in their core structures to stabilize chaperone-client interactions[35]. At this point, however, neither the precise arrangements of the N-terminal extensions in sHsps-client complexes are known, nor is it clear whether all clients use the same or client-specific binding interfaces[24,28]. In addition to these structural and activation-based similarities, mTXNPx and sHsps share also many mechanistic similarities; both proteins act as ATP-independent chaperone holdases on a wide range of unfolding client proteins, are able to transfer their client proteins after the heat shock treatment to ATP-dependent foldases for refolding and are essential for the heat shock survival of certain organisms[24,28]. Given the surprising fact that sHsps, which are highly conserved and ubiquitous, are largely absent from eukaryotic organelles, it is now tempting to speculate that mitochondrial 2-Cys-Prx might have taken over this crucial

function to protect the mitochondrial proteome during heat shock conditions not only in parasites but also in other eukaryotic organisms. Future studies are needed to ultimately determine whether the heat shock activated chaperone function of mitochondrial 2-Cys-Prx is parasite-specific or a general feature of mitochondrial 2-Cys-Prx.

## Methods

**Cryo-electron microscopy of mTXNPx$_{red}$–client complexes.** Mitochondrial mTXNPx without the mitochondrial targeting sequence (i.e., aa 1–26) but harboring a thrombin-cleavable 6-His-tag was purified from *E. coli* upon overexpression. Upon thrombin cleavage, the protein contains a 4-aa scar sequence (GSHM) on the N-terminus[14,15]. To prepare reduced mTXNPx, the protein was incubated with 5 mM DTT for 30 min at 30 °C. To allow complex formation between chaperone-active mTXNPx$_{red}$ (10 μM) and thermally unfolding luciferase (1 μM), the two proteins were incubated for 2 min at room temperature before they were slowly heated from 30 °C to 42 °C for 10 min[14]. Samples were collected after treatment at 42 °C and centrifuged at 16,100 × *g* for 30 min at 4 °C to remove large aggregates. For preparation of the cryo-EM samples, 2.5 μl of the mTXNPx:luciferase preparation or mTXNPx$_{red}$ alone, following heat activation, were applied to 1.2–1.3 μm holey Quantifoil grids at about 10 μM concentration. Grids were subsequently blotted for 2.5 s and plunge-frozen in liquid ethane using a Vitrobot Mark IV (FEI) at 8 °C and 100% humidity. Samples were imaged on a FEI Titan Krios (Thermo Fischer Scientific) operated at 300 kV at a magnification of 50,000× corresponding to a pixel size of 1.0 Å per pixel at the specimen level on a K2 summit direct electron detector (Gatan). A defocus range of 1.8–2.5 μm was used, with a total exposure of 8 s with 0.2 s subframes for a total of 50 e−/Å$^2$ accumulated dose and a dose rate of 1.25 e− per Å$^2$ per frame.

**Cryo-EM image processing.** Dose-fractionated image stacks were motion corrected using MotionCor2[12], discarding the first three frames. Motion corrected sums without dose weighting were used for contrast transfer function (CTF) estimation using GCTF[36]. Dose weighted, motion corrected sums were used for all subsequent processing. Reference based particle picking was done using Gautomatch (https://www.mrc-lmb.cam.ac.uk/kzhang/). For the mTXNPx-Luciferase dataset, a total of 286,068 particles were extracted from 2810 micrographs and subjected to reference-free 2D classification in Relion 2.1[37]. After removal of particles associated with noisy or contaminated classes, 213,512 particles remained corresponding to averages that showed high-resolution features in the decamer ring of mTXNPx and filled in the central lumen in top-down views, as well as all side-view averages. 3D classification resulted in models that had preferred orientation of either top or side views and was not used to further classify this dataset. Using a low pass filtered map generated from the *Leishmania braziliensis* mTXNPx crystal structure (4KB3) as an initial model, these particles were refined in Relion 2.1 with a soft mask and imposed D5 symmetry to yield a 3.7 Å resolution reconstruction after b-factor sharpening[38]. Without symmetry imposed, the mTXNPx-Luciferase dataset refined to 4.1 Å resolution. For the mTXNPx-apo dataset, a total of 593,338 particles were extracted from 2,368 micrographs which were reduced to a subset of 386,653 particles after both 2D and 3D classification in Relion 2.1 and removal of classes that did not help resolve the decamer structure. Refinement of these particles in Relion with a soft mask and imposed D5 symmetry resulted in a structure at 3.1 Å resolution. B-factors were estimated in Relion postprocessing and used to filter the final maps to their estimated resolution. Refinement in Relion followed gold-standard refinement procedures, and resolutions were estimated by the Fourier Shell Correlation (FSC) at 0.143[39]. Local resolution was estimated by the ResMap program[40] and suggested global maps varied in resolution from 3.7 to 4.1 Å in the mTXNPx-Luciferase dataset and from 2.9 to 3.3 Å in the mTXNPx$_{red}$ dataset. Model refinement was carried out using Rosetta[41] using the final maps generated in Relion as well as a homology model generated using the *Leishmania braziliensis* mTXNPx crystal structure (4KB3) and SWISS-MODEL[42]. Peptide backbone rebuilding and energy minimization was carried out in Rosetta using scripts based on the public scripts provided for refinement into EM density to perform local rebuilding in symmetric refinement. The resulting models were assessed and the N-terminal residues of mTXNPx were modeled in by hand in COOT[43]. After a final refinement procedure in PHENIX through Phenix.real_space_refine[44], both models' Ramachandran statistics were satisfactory as evaluated by MolProbity[45] and EMRinger[46].

**Generation of mTXNPx mutant proteins.** To introduce point mutations into mTXNPx, the full-length pET28c6His-THR-TmTXNPx plasmid, harboring the TmTXNPx sequence that codes for the mature protein (mTXNPx lacking the first 26 amino acids that compose the mitochondrial targeting peptide), was PCR-amplified with Pfu polymerase and primers containing the Y33A (5′- TCTGGAC TATCAGATG**GG**CCCGTACAGCGACTGTC-3′ and 5′- GACAGTCGCTGTACG G**GC**CATCTGATAGTCCAGA-3′) or the R34A (5′-GGACTATCAGATGTACG **C**TACAGCGACTGTCCGT-3′ and 5′- ACGGACAGTCGCTGTA**GC**GTACATC TGATAGTCC-3′) mutation (mutated nucleotides in bold). Upon digestion of parental DNA with 10 units of DpnI, the reactions were directly used to transform

*E. coli*. The accuracy of all constructs was verified by sequencing. The sequence encoding Δ5mTXNPx was PCR-amplified from pET28c6His-THR-TmTXNPx using *Pfu* polymerase and the primers 5′-ccgcgcacatatgTACCGTACAGCGACTGT CC-3′ and 5′-caccgctcgagTCACATGTTCTTCTCG AAAAAC-3′. The PCR fragment was subsequently digested with NdeI and XhoI for cloning into pET28c. Heterologous expression and purification of wild-type mTXNPx and mutants from *E. coli* BL21(DE3), as well as removal of the histidine tags with thrombin, were performed following previously described protocols[14,15]. Processed proteins were kept in 40 mM Hepes pH 7.5 and quantified by either $A_{280}$ measurements[14] or the bicinchoninic acid (BCA) protein assay, using bovine serum albumin (BSA) as standard.

**Chaperone activity assays.** The chaperone activity of purified recombinant, histidine tag-free mTXNPx (wild type and mutants) was measured by testing the effects of mTXNPx on the thermal aggregation of luciferase[14]. Briefly, mTXNPx was diluted into a cuvette containing 200 μM DTT in 40 mM Hepes pH 7.5. After a pre-incubation period of 5 min at 42 °C, 0.1 μM luciferase (Promega) was added to the reaction mixture and light scattering was measured under constant stirring using a fluorescence spectrophotometer Fluoromax-4 (Horiba) ($\lambda_{ex/em}$, 360 nm; slit widths, 2.5 nm). The light scattering signal of thermal luciferase in the absence of any chaperones was measured as control.

**Plasmid construction for in vivo expression.** To express mTXNPx without the mitochondrial import sequence or any additional tags, pET28c6His-THR-TmTXNPx[15] was PCR-amplified using Pfu polymerase and primers 5′-ccgcgca-catatgAATCTGGACTATCAGATGTAC-3′ and 5′-gcacatatgTATATCTCCTTCTT AAAGTTAAACAAAATTATTTCT-3′. The PCR product was digested with NdeI and subsequently re-ligated to originate a plasmid encoding mTXNPx devoid of any tag—pET28cTmTXNPx. The plasmid was further sequenced to ensure that no mutations on mTXNPx ORF had occurred and that the N-terminal 6-His-THR tag had been successfully removed.

**Production of mTXNPx—p-benzoyl-L-phenylalanine variants.** For the construction of mTXNPx-Bpa variants, we used a site-directed mutagenesis approach and substituted each phenylalanine or tyrosine codon in the sequence with an amber stop codon (TAG) using the primers listed in Supplementary Table 3. Following *Dpn*I digest of the parental DNA for 1 h at 37 °C and desalting, the PCR product was transformed into *E. coli* (XL1-Blue strain). Mutant colonies were sequenced and confirmed to carry the desired mutation.

**In vivo Bpa-mediated photo-cross-linking.** Bpa-mediated cross-linking was performed as previously described[47]. Briefly, *E. coli* cells (BL21 strain) were co-transformed with a plasmid encoding the respective mTXNPx-Bpa variant (pET28cTmTXNPx-Bpa) and a plasmid encoding the orthogonal aminoacyl-tRNA synthetase/tRNA pair (pEVOL) necessary for Bpa incorporation[20,21]. Cells were then plated in the presence of appropriate selective drugs and incubated overnight at 30 °C. Colonies were scraped off the plates and resuspended in LB medium ($OD_{600} = 0.4$), which contained a combination of antibiotics with or without 1 mM Bpa (Bachem AG, Bubendorf, Switzerland). Cells were then grown at 37 °C for 1 h. Afterwards, cells were cooled to 30 °C and expression of mTXNPx-Bpa variants and the tRNA synthetase/tRNA pair was induced with 10 μM IPTG and 0.1% L-arabinose, respectively. Cells were grown for 3 h at 30 °C, after which they were harvested and resuspended in PBS buffer using a volume of 100 μl per $OD_{600}$ of 1.0. Each sample was then subjected to a 30-min incubation at either 30 °C (control) or 45 °C (heat shock), transferred onto ice and exposed to a 10-min UV irradiation (366 nm) using a lamp distance of 2.5 cm. Cells were subsequently lysed using three cycles of sonication and two cycles of freeze-thaw. Total cell extracts were then supplemented with 5x SDS sample-loading buffer before analysis by SDS-PAGE. Immunoblotting using polyclonal anti-mTXNPx antibodies was used to visualize mTXNPx and its cross-linking products.

**In vitro cross-linking for LC-MS/MS analysis.** Purified mTXNPx[14] in either oxidized (mTXNPx$_{ox}$) or reduced forms (mTXNPX$_{red}$) (5 μM) and either alone or in complex with luciferase (0.5 μM) were incubated at 30 °C or 42 °C for 10 min. Samples were centrifuged at 16,100 ×g for 30 min at 4 °C. Supernatant was pipetted to a new microfuge tube and cross-linked using a 0.1 mM final concentration of DSA-$^{12}C_6$/$^{13}C_6$ or DSS-$H_{12}$/$D_{12}$ (equimolar mixes of isotopically light and isotopically heavy forms of the reagents; Creative Molecules Inc.). Cross-linking reactions were quenched by the addition of ammonium bicarbonate to a final concentration of 10 mM after a 15-min reaction time at RT. Samples were separated by reducing SDS/PAGE and portions of the gel corresponding to approximately 70–93 kDa (i.e., mTXNPx+/− luciferase) were each excised, subjected to in-gel digestion using trypsin (Promega), and subsequently prepared for LC-MS/MS analysis[48]. Quantitative cross-linking (qCL) experiments were performed in an identical fashion with respect to the reduction, temperature incubation, centrifugation, and cross-linking described above with the exception that no luciferase was added and that cross-linking was performed separately using either isotopically-light or isotopically-heavy forms of the cross-linking reagent for each condition. For both cross-linking reagents used (DSA and DSS), four distinct

sample conditions (mTXNPx$_{ox}$ at 30 °C, mTXNPx$_{ox}$ at 42 °C, mTXNPx$_{red}$ at 30 °C, or mTXNPx$_{red}$ at 42 °C) were prepared, split into two equal volumes, and cross-linked using a final concentration of 0.1 mM of the isotopically-light and isotopically-heavy forms of the cross-linking reagent, separately resulting in a total of eight cross-linked sample preparations per cross-linking reagent (DSA or DSS). These eight preparations were then combined in equal amounts according to the scheme described in Supplementary Figure 5B, ultimately yielding 16 unique combined samples (six forward-label and a corresponding six reverse-label samples) per cross-linking reagent. Combined samples were subsequently digested in-solution with trypsin and prepared for LC-MS/MS analysis as described[49].

**LC-MS/MS analysis.** Mass spectrometric analysis was performed using a nano-HPLC system (Easy-nLC II, ThermoFisher Scientific), coupled to the ESI-source of an LTQ Orbitrap Velos Pro (ThermoFisher Scientific). Samples were injected onto a 100 μm ID, 360 μm OD, 10 mm length trapping column packed with Magic C18AQ (Bruker-Michrom), 100 Å, 5 μm pore size (prepared in-house) and desalted by washing with Solvent A (2% acetonitrile: 98% water, both 0.1% formic acid (FA)). Peptides were separated with a 90-min gradient (mTXNPx + luciferase experiments: 0–90 min: 5–30% solvent B (90% acetonitrile, 10% water, 0.1% FA), 90–92 min: 30–100% B, 92–100 min: 100% B; mTXNPx qCL experiments: 0–90 min: 5–40% B, 90–92 min: 40–100% B, 92–100 min: 100% B), on a 75 μm ID, 360 μm OD, 150-mm length analytical column packed with Magic C18AQ 100 Å, 5 μm pore size (prepared in-house), with IntegraFrit (New Objective Inc.) and equilibrated with solvent A. MS data were acquired using a Top 8 data dependent method in which precursors ions with charge states ≥2 and a minimum signal intensity of 20,000 counts were considered for MS/MS acquisition. Dynamic exclusion was enabled with repeat count set to 1, exclusion mass widths relative to low or high each set to 10 ppm, and exclusion duration set to 60 s. MS and MS/MS events used 60,000 and 15,000 resolution FTMS scans, respectively, with a scan range of 400–2000 *m/z* in the MS. For MS/MS, CID collision energy was set to 35% and wideband activation was enabled.

**Cross-linking LC-MS/MS data analysis.** Cross-linking data was searched using Kojak (v.1.5.5)[50] against a concatenated target-decoy (reverse) sequence databases. All search output peptide-spectrum matches (PSMs) are included in Supplementary Data 1, 2. All cross-links that are discussed further in this text were required to be the single highest scoring PSM for their respective MS2 scan (i.e., Unique Score ID = "Unique") and to either have a score of 2 or greater (corresponding to an estimated FDR of 13.8%) or to have multiple corroborating PSMs with a score of at least 1 and have spectra manually inspected. Quantitation was performed on PSMs using XiQ (v.0.2)[51] (parameters: expected_shift = 0 and time_window = 1). The difference in cross-linking yield between experiments was expressed as "fold-change" and was calculated as the binary logarithm of the observed cross-link "Ratio H/L Median" as reported in the "tenpercent.csv" XiQ output. The fold-change for forward labeled experiments was calculated as the binary logarithm of the observed cross-link H/L ratio median as reported by XiQ. For the reverse-labeled experiments this was calculated as the negative binary logarithm of the observed cross-link H/L ratio (-log2(H/L)). Cross-link maps shown in Fig. 5a and Supplementary Figure 4B were generated using xiNET[52]. To facilitate interpretation of the in vitro LC-MS/MS cross-linking results, the identified cross-links were mapped onto the whole-chain model of luciferase (pdb: 1LCI). The MS datasets, search parameters and search outputs have been deposited to the ProteomeXchange Consortium via the PRIDE partner repository[53]. The accession number is: PXD010281.

**Oligomerization state of mTXNPx$_{red}$ in vitro.** To determine the oligomerization state of chaperone-active mTXNPx$_{red}$ (2 μM) either alone or in complex with 0.2 μM of luciferase at 30 °C or 42 °C, proteins were incubated with 40 mM GA for 10 min at the respective temperatures. Cross-linking was stopped by transferring 42 μl of each sample into a tube containing 8 μl of 1 M Tris-HCl, pH 8.0. The cross-linked species were separated by SDS-PAGE under reducing conditions and visualized by silver staining. The bands were quantified using the ImageJ software.

**Negative-stain EM.** Reaction mixtures of mTXNPx$_{red}$ (10 μM) either alone or in complex with 1 μM of luciferase at 30 °C or 42 °C were either immediately or after incubation at 4 °C applied onto thin carbon layered 400-mesh copper grids (Pelco) as described[54], negatively stained with 0.75% uranyl formate (pH 5.5–6.0) and imaged at room temperature using a a Tecnai T12 Microscope (FEI) equipped with a LaB$_6$ filament operated at 120 kV. Images were collected at 69444× magnification with a 2.2 Å/pixel spacing on a 4 k × 4 k CCD camera (Gatan). Particle images were selected EMAN2 software e2boxer on micrographs with similar stain thickness to estimate the number of decamers present in each micrograph.

## Data availability

The MS datasets, search parameters and search outputs have been deposited to the ProteomeXchange Consortium via the PRIDE partner repository. The accession number is: PXD010281. The structures have been deposited in the protein data base; mTXNPx

with bound client: 6E0F, EMD-8946, and apo mTXNPx: 6E0G, EMD-8947. Other data are available from the corresponding authors upon reasonable request.

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

## Acknowledgements

We thank Bastian Groitl for helpful discussions regarding in vivo cross-linking experiments and Peter Schultz for providing the plasmid encoding the aminoacyl-tRNA synthetase/suppressor tRNA pair. We thank Spencer Meeker for his assistance with in vitro

cross-linking experiments and Evgeniy Petrotchenko for his contribution of the isotopically-labelled cross-linking reagents. This work was supported by National Institute of Health Grants GM122506 to U.J. and R01GM110001 to D.S. and by Programa Operacional Regional do Norte [(ON.2—O Novo Norte under the Quadro de Referência Estratégico Nacional (QREN), through the Fundo Europeu de Desenvolvimento Regional (FEDER)] and FCT (Fundação para a Ciência e Tecnologia), co-funders of project NORTE-07–0124-FEDER-000002 to AMT through i3S. FCT provided funding to FT and HC under, respectively, the fellowship SFRH/BD/70438/2010 and the "Investigador FCT" contract IF/01244/2015. The University of Victoria-Genome British Columbia Proteomics Centre is supported by the Genome Canada and Genome British Columbia for operations (204PRO) and technology development (214PRO) through the Genome Innovations Network, and the Genomics Technology Platform (264PRO). C.H.B. is supported by the Leading Edge Endowment Fund (University of Victoria), the National Science and Engineering Research Council of Canada (NSERC), the Segal McGill Chair in Molecular Oncology at McGill University (Montreal, Quebec, Canada), the Warren Y. Soper Charitable Trust and the Alvin Segal Family Foundation to the Jewish General Hospital (Montreal, Quebec, Canada). B.A.M. and J.C.B. are funded by the Howard Hughes Medical Institute.

## Author contributions

F.T., E.T., K.M, H.C., and B.A.M.: Experimental design, data analysis, and manuscript preparation. C.B, A.M.T., L.P., and J.B. Data analysis, manuscript preparation D.S., U.J.: Study concept, data analysis, and manuscript preparation.

## Additional information

**Competing interests:** The authors declare no competing interests.

