## [Peer Review File · Nature Communications]

Reviewers' comments:

Reviewer #1 (Remarks to the Author):

Review of NCOMMS-18-20666-T 'Chaperone Activation and Client Binding of a 2-Cysteine Peroxiredoxin'

This is an interesting paper exploring structural rearrangements and substrate recognition by an unusual type of 2-Cys peroxiredoxin (mTXNPx) from the parasite *Leishmania infantum*. The conclusions proposed would add significantly to the emerging field of how 2-Cys peroxiredoxins could carry out their poorly understood chaperone function and likely alter the way researchers think around this topic.

In previous papers, reduced mTXNPx was shown to carry out a peroxidase-independent function preventing the aggregation of model proteins *in vitro* as well as stimulating the growth of the parasite at elevated temperatures (Texeira et al, PNAS, 2015). In addition, electron microscopical investigations suggested that the protein forms decameric ring structures that seem to coordinate an unfolded substrate at the center of the ring. This view is contrasting with one originating in studies on the structure of a chaperone-active species of another parasite 2 Cys peroxiredoxin (*Schistosoma mansoni* PrxI), that pointed to substrate binding at the outside of a double decameric chaperone species in part via residues in the C-terminus. Here the authors advance the field significantly by in greater detail resolving the mTXNPx structure in the presence of substrate and by complementary cross-linking studies.

In its present form the manuscript is a bit descriptive, however, lacking information on what is the importance of the substrate binding residues identified to the ability of mTXNPx to interact with/chaperone substrates. This and some technical issues need to be amended before the manuscript can be published.

Specific comments:

P6, line 25-26-p7 line 1: Based on the proposal that Y33 and R34 likely directly contact bound luciferase during mTXNPx chaperoning it would be interesting to know how amino acid-substitutions in any of these residues affects chaperone activity. A reduced substrate affinity in such mutant mTXNPx would support the claims made on the crucial initial substrate contacts performed by the N-terminus.

P8, lines 11-12: Based on the previous observations that mTXNPx is only activatable as a chaperone when in the reduced (decameric) form it is not clear why the authors choose only to evaluate the functionality of the Bpa-substituted alleles of mTXNPx in an assay relevant mostly for the independent peroxidase function (ie the *in vivo* oxidation redox status (dimerization), Fig. S2D).

A more relevant assay would be to assess to what extent the engineered Bpa-substituted alleles maintain chaperone activity (ie the ability to prevent aggregation of luciferase) or form reduced decamers (eg by native PAGE, gel filtration or glutaraldehyde cross-linking). This is necessary to evaluate to what extent the alleles used might be expected to report on native *in vivo* substrate interactions.

P14, lines 11-17: Results obtained using glutaraldehyde crosslinking need backup through quantification and an estimation of variation over multiple repetitions of the experiment.

In particular, it is not evident from Fig. 5 to what extent 'the presence of the *in vitro* client protein luciferase during the incubation process at 42°C appeared to stabilize mTXNPxred decamers'. In addition, I could not find glutaraldehyde cross-link data supporting the statement 'Subsequent cooling of the sample from 42°C to 30°C followed by cross-linking at 30°C re-established the decameric cross-links' in Fig. 5 as claimed in line 13-14.

Given that the authors propose that the protein becomes a dimer upon heat treatment additional support for the decamer-to-dimer transition might be obtained using different techniques eg native PAGE/gel filtration. Better support for this novel and crucial part of the paper is important to back

up the conclusions made.

On a final note, data to support the authors claim that mTXNPx_{red} decamer-to-dimer transition is essential for mTXNPx chaperone activity using specific point mutants (eg ones not able to destabilize the Cp-loop helix and conversely ones that constitutively destabilize it) would significantly support the author's model of the dimer-dimer interface playing a central role in chaperone activation and that 'destabilization of the Cp-loop helix is sufficient to turn these proteins into chaperones without the need for any other cues.' (p17, line 25-p18, line 2).

Reviewer #2 (Remarks to the Author):

In their manuscript "chaperone activation and client binding of a 2-Cysteine Peroxiredoxin" Teixeira et al. investigate a 2-Cysteine Peroxiredoxin. Many 2-Cys-peroxiredoxins are dual-function proteins that act as peroxidases under non-stress conditions or as chaperones during stress. Teixeira et al. now use the mitochondrial 2-Cys-Prx (mTXNPx) from *Leishmania infantum* to study its chaperone activity using cryo-EM and crosslinking MS. They were able to obtain a high-resolution structure of mTXNPx by Cryo-EM with and without bound model substrate luciferase and can demonstrate that the substrate gets bound in an unfolded state in the center of the decameric ring by the flexible N-termini of mTXNPx. The authors now use both in-vivo and in-vitro crosslinking and can convincingly show that the N-termini of mTXNPx are indeed involved in client binding. This is very compelling data which should be of high interest to the community and which clearly expands our molecular knowledge on how 2-cysteine peroxiredoxins are able to carry out their chaperone function.

The crosslinking data however implies that additional residues at the dimer-dimer interface, which when mapped onto the Cryo-EM structure are not in contact with the substrate protein, are involved in client binding. In a last set of experiments the authors then make use of quantitative crosslinking MS and crosslinking experiments using glutaraldehyde that suggest that elevated temperatures indeed trigger dissociation of the client free mTXNPx decamer, reminiscent of what is known from small heat-shock proteins.

This second part of the data is also highly interesting and novel, however, at the current state it is unclear how the Cryo-EM data, which excludes a major conformational rearrangement upon client binding, shall be reconciled with the crosslinking data.

While a full answer to this question is clearly beyond the scope of this paper – e.g. the small chaperone field has to my knowledge as well not being able to convincingly demonstrate that the active dimer actually is able to bind unfolding client proteins – the current manuscript raises some questions that should be addressed.

In their in-vivo crosslinking data Teixeira et al. see a reproducible smear of higher migrating species that are assigned by the authors to likely represent crosslinked complexes between TXNPx and client proteins. As it is a quite important point of this paper to establish which parts of TXNPx are involved in client binding, the authors should consider to cut-out those (or at least some select) bands and probe them by MS in order to see if they can detect client proteins as well as TXNPx peptides from those bands. They should also check this way at least once if the suspected dimer band is indeed from TXNPx.

The authors also find that the only two tyrosine residues that formed irreversible UV induced cross-links with other proteins independent of temperature were located in the N-terminus (Y30, Y33), which is in clear contradiction to the EM data, as the authors point out themselves. This is a quite puzzling result -the authors should discuss this a bit more extensively.

Lastly, the group of BPA variants that does not produce any higher migrating bands (group B3) – where are they located when mapped onto the solved cryo-EM structure? Does their location support the dimer-dissociation hypothesis?

Regarding the in-vitro crosslinking data, which also looks clear and compelling, but its presentation, at least to the reviewer, is sometimes a bit unclear.

E.g.: Two excel sheets are provided – “174529_0_supp_365863_pb1md3” and “174529_0_supp_365864_pb1md4” – are those the ones referred to in the Supplemental Material and the main text as “Table S3” and “Table S4”?

If so, why are then only intralinks within TXNPx that contain a Kojak Score ≥ 2 Uniq shown (second sheet in both excel files) and no TXNPx: Luciferase interlinks?

It is also interesting to see that there appears to be more inter than intralinks to be detected (Fig 4A and Fig S3B), which is rather unusual for an in-vitro crosslinking experiment, in particular as TXNPx is a decamer. Would this change if an even higher Kojak Score was applied? Perhaps the authors could comment on this.

And along the same lines:

How does the intralink pattern for TXNPx and luciferase look without the chaperone - e.g. lanes 2 to 7, Figure S3 – do they resemble the intralink patterns for the mTXNPxred + luciferase + CL sample – i.e. do they also support an unfolded luciferase?

Were Gel pieces for lane 8, mTXNPxox + luciferase + CL, also excised from 70-93 kDa as a control and analyzed with Kojak?

The name of one of the used crosslinkers DiSuccinimidylGlutarate on page 13 does not fit its acronym (DSS).

Reviewer #3 (Remarks to the Author):

The submitted manuscript by Teixeira and Tse et al-“Chaperone Activation and client binding of a 2-Cysteine Peroxiredoxin” reads well. It shows the cryo-EM structural of mTXNPxred as well as mTXNPxred-Luciferase complex in which they found that mTXNPxred forms a decameric ring. They have also shown that the decameric ring has the central cavity in which the partially folded Luciferase client interacts. Furthermore, they found critical residues of mTXNPxred which become exposed to solvent under high temperature. Additionally, they have also shown the critical residues of mTXNPxred implied in the client protein binding using cross-linking mass spectrometry.

After carefully reviewing the manuscript, I believe it does not add much to the existing knowledge in this field and there are some critical issues in this manuscript that need to be resolved. In its present state, I don't think it is up to the standard for publication in Nature Communications.

The biggest drawback of this paper is the fuzzy electron density of the Luciferase in the mTXNPxred-luciferase complex. Although the N-terminal interacting residues of mTXNPxred to Luciferase become ordered and the electron density on the figure looks good, the interacting residues of the Luciferase are nowhere to be seen. It could be due to fact that the 5-fold symmetry used during the data analysis has smeared the electron density for the client. A previous study from the same labs has shown an asymmetric mTXNPxred-luciferase complex in which Luciferase is more rigid and it is bound to a single dimer of mTXNPxred in the decameric ring. However, in the cryo-EM structure these asymmetric particles are not seen. The authors could try to use focused refinement, using a mask for Luciferase, to resolve its electron density without any symmetry application.

Although the structure of the decameric ring is of high resolution (and crystal structure of the analogue is also available without major conformational differences), the manuscript lacks information about the stoichiometry of the mTXNPxred-luciferase complex. It will be interesting to

see the refined structure without any applied symmetry to see how many luciferase molecules are bound to the ring. SEC (size exclusion chromatography) or Analytical Ultra Centrifugation (AUC) of the mTXNP_xred-luciferase complex would be useful. Additionally, intact mass-spectrometry could be helpful for the stoichiometry information as well as the assembly pathways of the mTXNP to the client.

Reviewer #1 (Remarks to the Author):

Review of NCOMMS-18-20666-T 'Chaperone Activation and Client Binding of a 2-Cysteine Peroxiredoxin'

This is an interesting paper exploring structural rearrangements and substrate recognition by an unusual type of 2-Cys peroxiredoxin (mTXNPx) from the parasite *Leishmania infantum*. The conclusions proposed would add significantly to the emerging field of how 2-Cys peroxiredoxins could carry out their poorly understood chaperone function and likely alter the way researchers think around this topic.

In previous papers, reduced mTXNPx was shown to carry out a peroxidase-independent function preventing the aggregation of model proteins *in vitro* as well as stimulating the growth of the parasite at elevated temperatures (Texeira et al, PNAS, 2015). In addition, electron microscopical investigations suggested that the protein forms decameric ring structures that seem to coordinate an unfolded substrate at the center of the ring. This view is contrasting with one originating in studies on the structure of a chaperone-active species of another parasite 2 Cys peroxiredoxin (*Schistosoma mansoni* PrxI), that pointed to substrate binding at the outside of a double decameric chaperone species in part via residues in the C-terminus. Here the authors advance the field significantly by in greater detail resolving the mTXNPx structure in the presence of substrate and by complementary cross-linking studies.

In its present form the manuscript is a bit descriptive, however, lacking information on what is the importance of the substrate binding residues identified to the ability of mTXNPx to interact with/chaperone substrates. This and some technical issues need to be amended before the manuscript can be published.

We thank the reviewer for his/her supportive comments.

Specific comments:

P6, line 25-26-p7 line 1: Based on the proposal that Y33 and R34 likely directly contact bound luciferase during mTXNPx chaperoning it would be interesting to know how amino acid-substitutions in any of these residues affects chaperone activity. A reduced substrate affinity in such mutant mTXNPx would support the claims made on the crucial initial substrate contacts performed by the N-terminus.

We thank the reviewer for this suggestion. We have now constructed a number of different mutants; one N-terminal truncation mutant, which lacks the first 5 aa, and two individual substitution mutants with either Y33 or R34 replaced with Alanine. We found that truncating the first 5 aa reduces the chaperone function of mTXNPx by over 70% (new Fig. 2A). Substitution of Y33 with the smaller hydrophobic Ala also caused a slight reduction in activity while substituting the charged R34 with a hydrophobic Ala massively increased the chaperone function (new Fig. 2B). The peroxidase activity of the variants was not significantly different indicating that the loss in chaperone activity is not due to pronounced structural alterations (new Fig. S2B). These results fit very well with our cryo-EM structure and confirm that the N-terminus is crucial for client binding. These results might also explain why cytosolic forms of TXNPx, which lack the N-terminal extensions, are not chaperone active *in vitro* (new Fig. S2C, D).

P8, lines 11-12: Based on the previous observations that mTXNPx is only activatable as a chaperone when in the reduced (decameric) form it is not clear why the authors choose only to evaluate the functionality of the Bpa-substituted alleles of mTXNPx in an assay relevant mostly for the independent peroxidase function (ie the *in vivo* oxidation redox status (dimerization), Fig. S2D). A more relevant assay would be to assess to what extent the engineered Bpa-substituted alleles maintain chaperone activity (ie the ability to prevent aggregation of luciferase) or form reduced decamers (eg by native

PAGE, gel filtration or glutaraldehyde cross-linking). This is necessary to evaluate to what extent the alleles used might be expected to report on native *in vivo* substrate interactions.

We attempted to purify several Bpa and tFpa (tri-fluorophenylalanine - a related structural analog used for NMR studies) mutants (i.e., Y30Bpa, Y73Bpa, Y73tFpa, F77Bpa, F79Bpa, Y111Bpa) but were unable to cleave the His-tag (which is only present for the purpose of purification) from the Y30Bpa, F77Bpa and F79Bpa mutant variant. However, we were able to purify and test the Y73Bpa, Y73tFpa and Y111Bpa mutants, which all showed wild-type-like chaperone activity (see attached Fig. 1, 2). Based on this result, the fact that Bpa is structurally very similar to Phe and Tyr, and the finding that all proteins can be expressed in a soluble form *in vivo*, we are confident that the proteins that we see crosslinking with the Bpa mutants are indeed specific client proteins. The reason why we tested the monomer-dimer status of the Bpa variants *in vivo* was to make sure that the variants are predominantly reduced and hence activatable *in vivo*.

Fig. 1. Influence of a 20-fold excess of wild-type or Y73tFpa mutant on the aggregation of thermally unfolding luciferase. Please see figure legend 2 in the main text for details

Fig. 2. Influence of a 20-fold excess of wild-type, Y73Bpa mutant or Y111 Bpa on the aggregation of thermally unfolding luciferase. Luciferase was incubated either alone or in the presence of reduced wt or mutant mTXNPx at 42°C for 10 min. After the incubation, the samples were spun down and the soluble supernatant was loaded onto a 14% SDS-PAGE. While most of the luciferase aggregates and disappears from the soluble supernatant, presence of wild-type and mutant mTXNPx maintains the solubility of the enzyme.

P14, lines 11-17: Results obtained using glutaraldehyde crosslinking need backup through quantification and an estimation of variation over multiple repetitions of the experiment.

In particular, it is not evident from Fig. 5 to what extent ‘the presence of the in vitro client protein luciferase during the incubation process at 42°C appeared to stabilize mTXNP_{x,red} decamers’. In addition, I could not find glutaraldehyde cross-link data supporting the statement ‘Subsequent cooling of the sample from 42°C to 30°C followed by cross-linking at 30°C re-established the decameric cross-links’ in Fig. 5 as claimed in line 13-14.

We have now repeated the complete set of experiments in triplicate. This was necessary since we realized that we did not perform the 43°C to 30°C shift experiments that the reviewer requested in replicates. These experiments were now done in the lab of our collaborator following the same protocol. As shown in our new Figure 6 A, B, the results are even clearer, presumably because we are able to quantitatively crosslink all decamers and dimers as illustrated by the complete absence of any monomers on our SDS-PAGE. We have conducted the experiments in triplicate and quantified the data (new Figure 6B). These results are in excellent agreement with our other data and demonstrate that the majority of decamers indeed dissociate into dimers upon shift to lower temperatures, and that presence of luciferase stabilizes the decamers.

Given that the authors propose that the protein becomes a dimer upon heat treatment additional support for the decamer-to-dimer transition might be obtained using different techniques eg native PAGE/gel filtration. Better support for this novel and crucial part of the paper is important to back up the conclusions made.

We agree that it would be great to have yet another method to demonstrate that the protein dissociates into dimers upon heat treatment. However, since this is a reversible event and re-association happens rapidly upon cooling down the protein, none of the suggested methods such as native PAGE/gel filtration can be used. We do think, however, by using three independent methods, i.e., GA-crosslinking, DSS-crosslinking and TEM, we have convincing evidence that temperature-induced dissociation occurs.

On a final note, data to support the authors claim that mTXNP_{x,red} decamer-to-dimer transition is essential for mTXNP_x chaperone activity using specific point mutants (eg ones not able to destabilize the Cp-loop helix and conversely ones that constitutively destabilize it) would significantly support the author’s model of the dimer-dimer interface playing a central role in chaperone activation and that ‘destabilization of the Cp-loop helix is sufficient to turn these proteins into chaperones without the need for any other cues.’ (p17, line 25-p18, line 2).

We also agree with this suggestion but feel that this aspect would go beyond the scope of this manuscript, and will constitute a separate study.

Reviewer #2 (Remarks to the Author):

In their manuscript “chaperone activation and client binding of a 2-Cysteine Peroxiredoxin” Teixeira et al. investigate a 2-Cysteine Peroxiredoxin. Many 2-Cys-peroxiredoxins are dual-function proteins that act as peroxidases under non-stress conditions or as chaperones during stress. Teixeira et al. now use the mitochondrial 2-Cys-Prx (mTXNP_x) from *Leishmania infantum* to study its chaperone activity using cryo-EM and crosslinking MS. They were able to obtain a high-resolution structure of mTXNP_x by Cryo-EM with and without bound model substrate luciferase and can demonstrate that the substrate gets bound in an unfolded state in the center of the decameric ring by the flexible N-termini of mTXNP_x. The authors now use both in-vivo and in-vitro crosslinking and can convincingly show that the N-termini of mTXNP_x are indeed involved in client binding. This is very compelling data which should be of high interest to the community and which clearly expands our molecular knowledge on how 2-cysteine peroxiredoxins are able to carry out their chaperone function.

We are very grateful for the positive and encouraging comments.

The crosslinking data however implies that additional residues at the dimer-dimer interface, which

when mapped onto the Cryo-EM structure are not in contact with the substrate protein, are involved in client binding. In a last set of experiments the authors then make use of quantitative crosslinking MS and crosslinking experiments using glutaraldehyde that suggest that elevated temperatures indeed trigger dissociation of the client free mTXNPx decamer, reminiscent of what is known from small heat-shock proteins. This second part of the data is also highly interesting and novel, however, at the current state it is unclear how the Cryo-EM data, which excludes a major conformational rearrangement upon client binding, shall be reconciled with the crosslinking data.

While a full answer to this question is clearly beyond the scope of this paper – e.g. the small chaperone field has to my knowledge as well not being able to convincingly demonstrate that the active dimer actually is able to bind unfolding client proteins – the current manuscript raises some questions that should be addressed.

In their in-vivo crosslinking data Teixeira et al. see a reproducible smear of higher migrating species that are assigned by the authors to likely represent crosslinked complexes between TXNPx and client proteins. As it is a quite important point of this paper to establish which parts of TXNPx are involved in client binding, the authors should consider to cut-out those (or at least some select) bands and probe them by MS in order to see if they can detect client proteins as well as TXNPx peptides from those bands. They should also check this way at least once if the suspected dimer band is indeed from TXNPx.

We agree that this would be very desirable but the issue is that we are loading cell lysate and visualize mTXNPx and crosslinked proteins by westernblot. Hence cutting out the bands would give us tremendous background signals. For the same reason that we are using highly specific anti-mTXNPx antibodies, we are confident that the dimer band is indeed mTXNPx.

The authors also find that the only two tyrosine residues that formed irreversible UV induced cross-links with other proteins independent of temperature were located in the N-terminus (Y30, Y33), which is in clear contradiction to the EM data, as the authors point out themselves. This is a quite puzzling result -the authors should discuss this a bit more extensively.

We agree that this is puzzling and have addressed this now in more detail.

Lastly, the group of BPA variants that does not produce any higher migrating bands (group B3) – where are they located when mapped onto the solved cryo-EM structure? Does their location support the dimer-dissociation hypothesis?

We have mapped those in the structure in the revised Fig. 4A and B. We found that these Group 3 (non-crosslinking) residues (i.e., Y145, Y160, Y192) clustered in the monomer-monomer B-type interface, which stabilizes the dimer.

Regarding the in-vitro crosslinking data, which also looks clear and compelling, but its presentation, at least to the reviewer, is sometimes a bit unclear.

E.g.: Two excel sheets are provided – “174529_0_supp_365863_pb1md3” and “174529_0_supp_365864_pb1md4” – are those the ones referred to in the Supplemental Material and the main text as “Table S3” and “Table S4”?

We apologize for the error. We have now attached the correct files.

If so, why are then only intralinks within TXNPx that contain a Kojak Score \geq 2_Uniq shown (second sheet in both excel files) and no TXNPx: Luciferase interlinks?

The reason why no *inter*-protein cross-links to luciferase are shown because this workbook contains the results for the qCL experiments, in which we worked only with mTXNPx in the absence of luciferase. We have now uploaded the correct Table S3, in which you will see all PSMs listed (mTXNPx:mTXNPx and Luciferase:Luciferase intra-protein cross-links as well as mTXNPx:Luciferase inter-protein crosslinks.

It is also interesting to see that there appears to be more inter than intralinks to be detected (Fig 4A

and Fig S3B), which is rather unusual for an in-vitro crosslinking experiment, in particularly as TXNPx is a decamer. Would this change if an even higher Kojak Score was applied? Perhaps the authors could comment on this.

We apologize for the confusion: Figure S3B (now Figure S4B) depicts crosslinks on a luciferase monomer. This fits well with our stoichiometry data, which showed that only one luciferase molecule is bound per decamer. We have clarified this point.

Figure legend S4B. “Crosslinking pattern of thermally unfolded luciferase in complex with *mTXNPx_{red}*. *mTXNPx_{red}*– luciferase complexes were formed, crosslinked with DSS and excised (Fig. S3A, lane 9). After tryptic digestion, the peptides were analyzed by MS/MS analysis. All crosslinks were intra-crosslinks within the luciferase monomer. Short-range (purple) and long range (yellow, red, cyan) cross-links are indicated on the linear structure (upper panel) or on the crystal structure (pdb 1LCI) of firefly luciferase. Results with both cross-linkers are shown (Table S3).”

Regarding more unique inter than intralinks being depicted in Figure 4A. This may have to do with the fact that Luciferase may bind in various orientations, hence differentially interacting with the decamer chaperone. This would increase the possible number of interlinks that could be formed/observed and may, in part, explain this skew toward more interlinks being observed.

Would this change if an even higher Kojak Score was applied? Perhaps the authors could comment on this.

It appears to us that higher score thresholds do not affect the inter/intralink distribution in any interesting way. The reviewer can evaluate this using the score slider

here: <http://crosslinkviewer.org/uploaded.php?uid=0d249101b41da624cfd28a86b1d9acebc93cfa25>.

And along the same lines:

How does the intralink pattern for TXNPx and luciferase look without the chaperone - e.g. lanes 2 to 7, Figure S3 – do they resemble the intralink patterns for the *mTXNPx_{red}* + luciferase + CL sample – i.e. do they also support an unfolded luciferase? Were Gel pieces for lane 8, *mTXNPx_{ox}* + luciferase + CL, also excised from 70-93 kDa as a control and analyzed with Kojak?

Unfortunately, we do not have that data.

The name of one of the used crosslinkers DiSuccinimidylGlutarate on page 13 does not fit its acronym (DSS).

We fixed this error. It should read “DiSuccinimidylSuberate”

Reviewer #3 (Remarks to the Author):

The submitted manuscript by Teixeira and Tse et al-“Chaperone Activation and client binding of a 2-Cysteine Peroxiredoxin” reads well. It shows the cryo-EM structural of *mTXNPx_{red}* as well as *mTXNPx_{red}*-Luciferase complex in which they found that *mTXNPx_{red}* forms a decameric ring. They have also shown that the decameric ring has the central cavity in which the partially folded Luciferase client interacts. Furthermore, they found critical residues of *mTXNPx_{red}* which become exposed to solvent under high temperature. Additionally, they have also shown the critical residues of *mTXNPx_{red}* implied in the client protein binding using cross-linking mass spectrometry.

We thank the reviewer for the positive evaluation.

After carefully reviewing the manuscript, I believe it does not add much to the existing knowledge in this field and there are some critical issues in this manuscript that need to be resolved. In its present state, I don't think it is up to the standard for publication in Nature Communications.

1. The biggest drawback of this paper is the fuzzy electron density of the Luciferase in the *mTXNPx_{red}*-luciferase complex. Although the N-terminal interacting residues of *mTXNPx_{red}* to

Luciferase become ordered and the electron density on the figure looks good, the interacting residues of the Luciferase are nowhere to be seen. It could be due to fact that the 5-fold symmetry used during the data analysis has smeared the electron density for the client. A previous study from the same labs has shown an asymmetric mTXNPx_{red}-luciferase complex in which Luciferase is more rigid and it is bound to a single dimer of mTXNPx_{red} in the decameric ring. However, in the cryo-EM structure these asymmetric particles are not seen. The authors could try to use focused refinement, using a mask for Luciferase, to resolve its electron density without any symmetry application.

This was an excellent suggestion and we have since performed a focused refinement using a mask in the central region of the complex that includes the bound luciferase and the N-terminal strands. These results have been added to Supplemental Fig. S1J and discussed in the results (p. 7). Unfortunately, in both symmetrized and unsymmetrized maps the density for luciferase was not ordered enough to resolve its structure or the regions that interact with the mTXNPx_{red} N-terminal strands. Although, the N-terminal strands mTXNPx_{red} remained well-resolved for these refinements, indicating the refinement method worked. Additionally, the C1, unsymmetrized refinement of the entire complex (without masking) also did not resolve luciferase structure but did identify the same mTXNPx_{red} decamer arrangement at a lower resolution (Supplementary Fig. 1I). Additionally, in our reference-free 2D class averages (Figure 1A and Supplementary Fig. 1A), in which no symmetry is imposed, density for luciferase is not structured, while the mTXNPx_{red} decamer is well-resolved. Based on these results we conclude that luciferase is bound heterogeneously in the decamer ring and likely in an unstructured state, thus the interacting residues are not resolvable by single particle cryo-EM. This is not surprising considering that, under these conditions, binding by mTXNPx_{red} occurs following thermal unfolding of luciferase. Furthermore, in our previously published work (Teixeira, et al., PNAS, 2015, Figure 2A) we showed that luciferase fluorescence is fully restored from thermal unfolding only after the addition of the KJE chaperone system (DnaK, DnaJ and GrpE) in addition to mTXNPx. Thus, under the conditions in which the cryo-EM structures were determined, mTXNPx_{red} is likely stabilizing luciferase in an unstructured intermediate that is inactive but competent for refolding and activation in the presence of the KJE chaperone system.

Additionally, the reviewer points out that *“it will be interesting to see the refined structure without any applied symmetry to see how many luciferase molecules are bound to the ring”*. As suggested, we have included an asymmetric refinement (Supplementary Fig. S1I), however, as discussed above, luciferase appears to be bound in an unstructured intermediate state, thus by cryo-EM we cannot definitively assess the stoichiometry. Based on the 63 kDa size of luciferase and the volume of the central density in the structure, we estimate that a single luciferase molecule could appropriately fit in the density. However, given the heterogeneity, we cannot exclude the possibility of multiple luciferase molecules are bound in certain dodecamers. We have also determined the fraction of bound vs. unbound decamers in the dataset and identified that over 60% of mTXNPx_{red} are bound to luciferase. These data have been added to Supplementary Fig 1B and discussion of the above points has been added to the Results (p. 6).

2. Although the structure of the decameric ring is of high resolution (and crystal structure of the analogue is also available without major conformational differences), the manuscript lacks information about the stoichiometry of the mTXNPx_{red}-luciferase complex. It will be interesting to see the refined structure without any applied symmetry to see how many luciferase molecules are bound to the ring. SEC (size exclusion chromatography) or Analytical Ultra Centrifugation (AUC) of the mTXNPx_{red}-luciferase complex would be useful. Additionally, intact mass-spectrometry could be helpful for the stoichiometry information as well as the assembly pathways of the mTXNP to the client.

We thank the reviewer for the suggestion. To address this question, we conducted analytic ultracentrifugation experiments of complexes formed between mTXNPx and luciferase (new Figure S1E). Although our occupancy is not 100% (presumably due to fact that the AUC experiments take a long time), we clearly observe a 1:1 decamer:luciferase stoichiometry. Based on these experiments, we conclude that one decamer binds one unfolded luciferase molecule, a conclusion that also fits to the observed dimensions of the density in the center of the decamer (i.e., 63 kDa).

REVIEWERS' COMMENTS:

Reviewer #1 (Remarks to the Author):

Review of NCOMMS-18-20666-T 'Chaperone Activation and Client Binding of a 2-Cysteine Peroxiredoxin'

This is an interesting paper exploring structural rearrangements and substrate recognition by an unusual type of 2-Cys peroxiredoxin (mTXNPx) from the parasite *Leishmania infantum*. The conclusions proposed would add significantly to the emerging field of how 2-Cys peroxiredoxins could carry out their poorly understood chaperone function and likely alter the way researchers think around this topic.

In previous papers, reduced mTXNPx was shown to carry out a peroxidase-independent function preventing the aggregation of model proteins *in vitro* as well as stimulating the growth of the parasite at elevated temperatures (Texeira et al, PNAS, 2015). In addition, electron microscopical investigations suggested that the protein forms decameric ring structures that seem to coordinate an unfolded substrate at the center of the ring. This view is contrasting with one originating in studies on the structure of a chaperone-active species of another parasite 2 Cys peroxiredoxin (*Schistosoma mansoni* PrxI), that pointed to substrate binding at the outside of a double decameric chaperone species in part via residues in the C-terminus. Here the authors advance the field significantly by in greater detail resolving the mTXNPx structure in the presence of substrate and by complementary cross-linking studies.

In its present form the manuscript is a bit descriptive, however, lacking information on what is the importance of the substrate binding residues identified to the ability of mTXNPx to interact with/chaperone substrates. This and some technical issues need to be amended before the manuscript can be published.

I believe that the authors have convincingly addressed my concerns and, provided that the minor comments mentioned below are taken care of, think that the manuscript is now suitable for publication in Nature Communications.

We thank the reviewer for his/her supportive comments.

Specific comments:

P6, line 25-26-p7 line 1: Based on the proposal that Y33 and R34 likely directly contact bound luciferase during mTXNPx chaperoning it would be interesting to know how amino acid-substitutions in any of these residues affects chaperone activity. A reduced substrate affinity in such mutant mTXNPx would support the claims made on the crucial initial substrate contacts performed by the N- terminus.

We thank the reviewer for this suggestion. We have now constructed a number of different mutants; one N-terminal truncation mutant, which lacks the first 5 aa, and two individual substitution mutants with either Y33 or R34 replaced with Alanine. We found that truncating the first 5 aa reduces the chaperone function of mTXNPx by over 70% (new Fig. 2A). Substitution of Y33 with the smaller hydrophobic Ala also caused a slight reduction in activity while substituting the charged R34 with a hydrophobic Ala massively increased the chaperone function (new Fig. 2B). The peroxidase activity of the variants was not significantly different indicating that the loss in chaperone activity is not due to pronounced structural alterations (new Fig. S2B). These results fit very well with our cryo-EM structure and confirm that the N-terminus is crucial for client binding. These results might also explain why cytosolic forms of TXNPx, which lack the N-terminal extensions, are not chaperone active *in vitro* (new Fig. S2C, D).

The chaperone activity data of the mutants/cytosolic mTXNPx added taken together with the cryoEM data strongly supports the authors claims on the N-terminus playing a crucial role in

substrate interactions, congratulations! However, in my view the $\Delta 5$ and Y33A mutants seem to be slightly deficient in peroxidase function and therefore statements describing this in a more appropriate way are needed.

The way that the authors describe the data on p8, line 17: '...this N-terminally truncated mutant variant showed near wild-type like peroxidase activity...' is fine whereas a description more appropriate on p9, line 8 would be: '...As observed before, these mutations only mildly (Y33A) or not significantly (R34A) altered...'

P8, lines 11-12: Based on the previous observations that mTXNPx is only activatable as a chaperone when in the reduced (decameric) form it is not clear why the authors choose only to evaluate the functionality of the Bpa-substituted alleles of mTXNPx in an assay relevant mostly for the independent peroxidase function (ie the in vivo oxidation redox status (dimerization), Fig. S2D). A more relevant assay would be to assess to what extent the engineered Bpa-substituted alleles maintain chaperone activity (ie the ability to prevent aggregation of luciferase) or form reduced decamers (eg by native PAGE, gel filtration or glutaraldehyde cross-linking). This is necessary to evaluate to what extent the alleles used might be expected to report on native in vivo substrate interactions.

We attempted to purify several Bpa and tFpa (tri-fluorophenylalanine - a related structural analog used for NMR studies) mutants (i.e., Y30Bpa, Y73Bpa, Y73tFpa, F77Bpa, F79Bpa, Y111Bpa) but were unable to cleave the His-tag (which is only present for the purpose of purification) from the Y30Bpa, F77Bpa and F79Bpa mutant variant. However, we were able to purify and test the Y73Bpa, Y73tFpa and Y111Bpa mutants, which all showed wild-type-like chaperone activity (see attached Fig. 1, 2). Based on this result, the fact that Bpa is structurally very similar to Phe and Tyr, and the finding that all proteins can be expressed in a soluble form in vivo, we are confident that the proteins that we see crosslinking with the Bpa mutants are indeed specific client proteins. The reason why we tested the monomer-dimer status of the Bpa variants in vivo was to make sure that the variants are predominantly reduced and hence activatable in vivo.

Fig. 1. Influence of a 20-fold excess of wild-type or Y73tFpa mutant on the aggregation of thermally unfolding luciferase. Please see figure legend 2 in the main text for details

Fig. 2. Influence of a 20-fold excess of wild-type, Y73Bpa mutant or Y111 Bpa on the aggregation of thermally unfolding luciferase. Luciferase was incubated either alone or in the presence of reduced wt or mutant mTXNPx at 42°C for 10 min. After the incubation, the samples were spun down and the soluble supernatant was loaded onto a 14% SDS-PAGE. While most of the luciferase aggregates and disappears from the soluble supernatant, presence of wild-type and mutant mTXNPx maintains the solubility of the enzyme.

I believe the authors conclusion that the Bpa experiment reports on in vivo relevant interactions is correct but I think that the data presented in Fig. 2 for reviewers above would be valuable also to readers of Nature Communications. This since seeing near wild-type ability of at least some of the Bpa mutants constructed to keep luciferase native directly more reports on chaperone activity than analysis of in vivo redox state. Therefore, I suggest that the authors add the data as Fig S3E and discuss it appropriately in the text eg after p11, line 5.

P14, lines 11-17: Results obtained using glutaraldehyde crosslinking need backup through quantification and an estimation of variation over multiple repetitions of the experiment. In particular, it is not evident from Fig. 5 to what extent 'the presence of the in vitro client protein luciferase during the incubation process at 42°C appeared to stabilize mTXNPxred decamers'. In addition, I could not find glutaraldehyde cross-link data supporting the statement 'Subsequent cooling of the sample from 42°C to 30°C followed by cross-linking at 30°C re-established the decameric cross-links' in Fig. 5 as claimed in line 13-14.

We have now repeated the complete set of experiments in triplicate. This was necessary since we

realized that we did not perform the 43°C to 30°C shift experiments that the reviewer requested in replicates. These experiments were now done in the lab of our collaborator following the same protocol. As shown in our new Figure 6 A, B, the results are even clearer, presumably because we are able to quantitatively crosslink all decamers and dimers as illustrated by the complete absence of any monomers on our SDS-PAGE. We have conducted the experiments in triplicate and quantified the data (new Figure 6B). These results are in excellent agreement with our other data and demonstrate that the majority of decamers indeed dissociate into dimers upon shift to lower temperatures, and that presence of luciferase stabilizes the decamers.

Given that the authors propose that the protein becomes a dimer upon heat treatment additional support for the decamer-to-dimer transition might be obtained using different techniques eg native PAGE/gel filtration. Better support for this novel and crucial part of the paper is important to back up the conclusions made.

We agree that it would be great to have yet another method to demonstrate that the protein dissociates into dimers upon heat treatment. However, since this is a reversible event and re-association happens rapidly upon cooling down the protein, none of the suggested methods such as native PAGE/gel filtration can be used. We do think, however, by using three independent methods, i.e., GA-crosslinking, DSS-crosslinking and TEM, we have convincing evidence that temperature- induced dissociation occurs.

On a final note, data to support the authors claim that mTXNPxred decamer-to-dimer transition is essential for mTXNPx chaperone activity using specific point mutants (eg ones not able to destabilize the Cp-loop helix and conversely ones that constitutively destabilize it) would significantly support the author's model of the dimer-dimer interface playing a central role in chaperone activation and that 'destabilization of the Cp-loop helix is sufficient to turn these proteins into chaperones without the need for any other cues.' (p17, line 25-p18, line 2). We also agree with this suggestion but feel that this aspect would go beyond the scope of this manuscript, and will constitute a separate study.

Reviewer #2 (Remarks to the Author):

My comments from the first round of revision have been addressed satisfactorily by the authors.

Reviewer #3 (Remarks to the Author):

The Fuzzy electron density of the luciferase on the ring, data refinement without symmetry and the stoichiometry of the mTXNPx : luciferase complex

The above were my concerns and the authors applied focus refinement for the ring and the luciferase separately.

Previously I believed that the whole data set was biased by the presence of the higher order symmetry of the ring which left the client disordered. After focused refinement, it seems like the luciferase is indeed in a very unfolded state and it could not be resolved structurally. I am happy with the efforts authors made to refine the structure again to find that the more homogeneous electron density belongs to luciferase, even if they could not find it to answer my question. However, it could be a matter of in-vitro reconstitution of the complex which leads to this heterogeneity and this is beyond the scope for this manuscript. Although the authors could not define the stoichiometry of the complex based on the cryo-EM structure of the complex, it is addressed using AUC experiments and they found 1:1 stoichiometry of the complex.

I am happy to suggest the acceptance of the revised manuscript.

We thank the reviewers for critically reviewing our revised ms. We have made the suggested changes and responded to the critiques in the point-by-point response.

Reviewer 1: The color code is as following:

Green: Reviewers comments to first revision

Pink: Authors second response

Reviewer #1 (Remarks to the Author):

I believe that the authors have convincingly addressed my concerns and, provided that the minor comments mentioned below are taken care of, think that the manuscript is now suitable for publication in *Nature Communications*.

We thank the reviewer for his/her supportive comments.

The chaperone activity data of the mutants/cytosolic mTXNPx added taken together with the cryoEM data strongly supports the authors claims on the N-terminus playing a crucial role in substrate interactions, congratulations! However, in my view the $\Delta 5$ and Y33A mutants seem to be slightly deficient in peroxidase function and therefore statements describing this in a more appropriate way are needed.

The way that the authors describe the data on p8, line 17: '...this N-terminally truncated mutant variant showed near wild-type like peroxidase activity...' is fine whereas a description more appropriate on p9, line 8 would be: '...As observed before, these mutations only mildly (Y33A) or not significantly (R34A) altered...'

We have changed the sentence accordingly.

I believe the authors conclusion that the Bpa experiment reports on *in vivo* relevant interactions is correct but I think that the data presented in Fig. 2 for reviewers above would be valuable also to readers of *Nature Communications*. This since seeing near wild-type ability of at least some of the Bpa mutants constructed to keep luciferase native more directly reports on chaperone activity than analysis of *in vivo* redox state. Therefore, I suggest that the authors add the data as Fig S3E and discuss it appropriately in the text eg after p11, line 5.

As suggested by the reviewer, we have added the Fig.2 for reviewers as panel Fig. S3E and changed the text of the ms accordingly.

Reviewer #2 (Remarks to the Author):

My comments from the first round of revision have been addressed satisfactorily by the authors.

We thank the reviewer for his/her supportive comments

Reviewer #3 (Remarks to the Author):

The Fuzzy electron density of the luciferase on the ring, data refinement without symmetry and the stoichiometry of the mTXNPx : luciferase complex. The above were my concerns and the authors applied focus refinement for the ring and the luciferase separately.

Previously I believed that the whole data set was biased by the presence of the higher order symmetry of the ring which left the client disordered. After focused refinement, it seems like the luciferase is indeed in a very unfolded state and it could not be resolved structurally. I am happy with the efforts authors made to refine the structure again to find that the more homogeneous electron density belongs to luciferase, even if they could not find it to answer my question. However, it could be a matter of in-vitro reconstitution of the complex which leads to this heterogeneity and this is beyond the scope for this manuscript. Although the authors could not define the stoichiometry of the complex based on the cryo-EM structure of the complex, it is addressed using AUC experiments and they found 1:1 stoichiometry of the complex.

I am happy to suggest the acceptance of the revised manuscript.

We thank the reviewer for his/her supportive comments